# Large-scale genomic rearrangements boost SCRaMbLE in *Saccharomyces cerevisiae*

Li Cheng [1,12], Shijun Zhao[1,2,12], Tianyi Li[1,11,12], Sha Hou[1,12], Zhouqing Luo[1,3], Jinsheng Xu[4], Wenfei Yu [1,2], Shuangying Jiang [1], Marco Monti [5], Daniel Schindler[5], Weimin Zhang [6], Chunhui Hou [7], Yingxin Ma[1], Yizhi Cai [1,5], Jef D. Boeke [6,8] & Junbiao Dai [1,2,9,10] ✉

Synthetic Chromosome Rearrangement and Modification by LoxP-mediated Evolution (SCRaMbLE) is a promising tool to study genomic rearrangements. However, the potential of SCRaMbLE to study genomic rearrangements is currently hindered, because a strain containing all 16 synthetic chromosomes is not yet available. Here, we construct SparLox83R, a yeast strain containing 83 loxPsym sites distributed across all 16 chromosomes. SCRaMbLE of SparLox83R produces versatile genome-wide genomic rearrangements, including inter-chromosomal events. Moreover, when combined with synthetic chromosomes, SCRaMbLE of hetero-diploids with SparLox83R leads to increased diversity of genomic rearrangements and relatively faster evolution of traits compared to hetero-diploids only with wild-type chromosomes. Analysis of the SCRaMbLEd strain with increased tolerance to nocodazole demonstrates that genomic rearrangements can perturb the transcriptome and 3D genome structure and consequently impact phenotypes. In summary, a genome with sparsely distributed loxPsym sites can serve as a powerful tool for studying the consequence of genomic rearrangements and accelerating strain engineering in *Saccharomyces cerevisiae*.

Genomic rearrangements alter genetic linkage of discrete chromosomal regions and are an important mutational force in evolution[1,2], underlying the pathology of various germline and somatic diseases such as nervous system disorders[3], congenital heart diseases[4], and cancers[5]. The profound effects of these genomic rearrangements, which include deletions, duplications, insertions, inversions, and translocations, are attributed to alterations in gene copy numbers, disruption of protein-coding sequences, and perturbation of cis-regulatory networks[6–10]. Previous studies indicate that the rate of genomic rearrangements is several orders of magnitude higher than

[1]Shenzhen Key Laboratory of Synthetic Genomics, Guangdong Provincial Key Laboratory of Synthetic Genomics, Key Laboratory of Quantitative Synthetic Biology, Shenzhen Institute of Synthetic Biology, Shenzhen Institute of Advanced Technology, Chinese Academy of Sciences, Shenzhen 518055, China. [2]University of Chinese Academy of Sciences, Beijing, China. [3]State Key Laboratory of Cellular Stress Biology, Innovation Center for Cell Signaling Network, School of Life Sciences, Xiamen University, Xiamen, Fujian 361102, China. [4]Department of Bioinformatics, Huazhong Agricultural University, Wuhan 430070, China. [5]Manchester Institute of Biotechnology, University of Manchester, Manchester M1 7DN, UK. [6]Institute for Systems Genetics and Department of Biochemistry and Molecular Pharmacology, NYU Langone Health, New York, NY, USA. [7]China State Key Laboratory of Genetic Resources and Evolution, Kunming Institute of Zoology, Chinese Academy of Sciences, Kunming 650223, China. [8]Department of Biomedical Engineering, NYU Tandon School of Engineering, Brooklyn, NY 11201, USA. [9]Shenzhen Branch, Guangdong Laboratory for Lingnan Modern Agriculture, Key Laboratory of Synthetic Biology, Ministry of Agriculture and Rural Affairs, Agricultural Genomics Institute at Shenzhen, Chinese Academy of Agricultural Sciences, Shenzhen, China. [10]College of Life Sciences and Oceanography, Shenzhen University, 1066 Xueyuan Rd, Shenzhen 518055 Guangdong, China. [11]Present address: Shenzhen Lianghe Biotechnology Co., Ltd., Shenzhen, China. [12]These authors contributed equally: Li Cheng, Shijun Zhao, Tianyi Li, Sha Hou. ✉e-mail: daijunbiao@caas.cn

the rate of base substitutions[11], and genomic rearrangements are pervasive in all domains of life, from prokaryotes to humans[12].

Genomic rearrangements likely occur randomly and sporadically, and discriminating their impacts from the confounding effects of other mutation types remains a challenge[13,14]. To quantify the impact of genomic rearrangement independently, several methods have been developed to construct targeted long-range rearrangements in model organisms, including Cre/loxP recombination[13], I-SceI-induced[15] and CRISPR/Cas9-induced[14] DSB repair. However, only limited inter-chromosomal rearrangements can be generated and analyzed by these methods. A controlled system allowing genome-wide induction of a wide array of chromosomal rearrangements has not yet been established.

In the synthetic yeast genome project (Sc2.0), thousands of loxPsym sequences are incorporated in a designer genome to create a system known as Synthetic Chromosome Recombination and Modification by LoxP-mediated Evolution (SCRaMbLE)[16]. Through SCRaMbLE, massive rearrangements can be introduced into the synthetic chromosomes, including intra-chromosomal deletions, inversions, duplications, and inter-chromosomal translocations[17,18]. SCRaMbLE therefore provides a strategy to investigate the potential mechanisms of genomic rearrangements and their functional consequences and facilitates delineation of the relationships between large-scale genomic rearrangements and diseases. However, to date, the power of SCRaMbLE has mostly been used in single synthetic chromosome contexts[18–30]. Moreover, intra-chromosomal rearrangements dominate, with inter-chromosomal rearrangements rarely detected in individual SCRaMbLEd strains despite the utilization of a strain with four synthetic chromosomes[31]. By sequencing SCRaMbLEd pools derived from inducing a strain harboring 5.5 synthetic chromosomes, comparable frequencies of intra- and inter-chromosomal events were detected at 47.24% and 52.67%, respectively[32]. However, only 9.87% of the total reads accounted for inter-chromosomal recombination, indicating that the occurrence of inter-chromosomal recombination was significantly lower than that of intra-chromosomal recombination in cells containing multiple synthetic chromosomes. Considering that the spatial proximity between loxPsym sites plays a crucial role in chromosome interactions, it can be inferred that translocation events would be more frequent when loxPsym sites are distributed throughout the entire genome.

Here, CRISPR/Cas9 was used to simultaneously integrate loxPsym sequences into multiple genomic loci. The resultant extensively engineered yeast strain, SparLox83R, possessed 83 Sparsely distributed LoxPsym sites across the genome with ReSCuES system. After inducing expression of Cre recombinase, the resultant SCRaMbLEd yeast population underwent diverse large-scale genomic rearrangements, dominated by inter-chromosomal events. By quantifying rearrangement frequencies at loxPsym sites, this study revealed multiple factors impacting loxPsym-mediated genomic rearrangements, including chromatin accessibility and genomic distance between sites. The impacts of large-scale genomic rearrangements on cell fitness under stress were investigated in progeny populations, and translocation and duplication events were detected that led to increased tolerance of nocodazole. SparLox83R was further enhanced with the SCRaMbLE system by crossing with a haploid strain containing synIII (JDY541) from the Sc2.0 project. Inducing SCRaMbLE in this heterozygous diploid strain resulted in complex genomic rearrangements, including numerous loss of heterozygosity (LOH) and aneuploidy events, as well as rapid adaptation of tolerance to acetic acid (HAc). The loxPsym sites in SparLox83R support the rapid development of new phenotypes by substantially boosting the SCRaMbLE system with large-scale genomic rearrangements.

## Results

### Genome-wide insertion of 83 loxPsym sites throughout the yeast genome

Cre/LoxP-based SCRaMbLE cannot yet be employed in the Sc2.0 project across all 16 yeast chromosomes because construction of a fully synthetic yeast genome is still underway. Previous studies employing SCRaMbLE in strains with multiple synthetic chromosomes reported the frequency of inter-chromosomal rearrangements as less than 10%[31,32]. To construct an engineered yeast strain suitable for effective introduction of genome-wide, large-scale genomic rearrangements, loxPsym-containing sequences were introduced by design into all the yeast chromosomes using a CRISPR-Cas9-based gene editing strategy (Supplementary Fig. 1a). All the insertion sites were designed at the intergenic regions which contained a PAM site, NGG, without disrupting any possible functional elements. Briefly, three loci were simultaneously targeted for integration of loxPsym sites in each cycle, eventually producing a strain with 83 sites (SparLox83) distributed across the 16 chromosomes, with at least two sites per chromosome (Fig. 1a). The presence of loxPsym at each locus was confirmed by whole-genome sequencing (WGS) (Supplementary Data 1, 2). The final strain exhibited four off-target insertions, namely II-3, II-8, IV-8, and V-2. Among these sites, three (II-3, II-8, and V-2) were located within the ORFs of non-essential genes. Since the strain did not display any discernible growth defects (Fig. 1b), we opted to retain the three loxPsym sites. Meanwhile, multiple loxPsym sites were identified at the same loci through self-duplication of the plasmid backbone sequence (Fig. 1a and Supplementary Fig. 1b). LoxPsym sites in SparLox83 were sparsely distributed, with significantly longer genomic distances between adjacent sites than in Sc2.0 synthetic chromosomes. The average genomic distance between adjacent loxPsym sites was ~145 kb in SparLox83, much larger than the 3 kb distance between sites in Sc2.0 chromosomes (Supplementary Fig. 1c). Furthermore, WGS and pulsed-field gel electrophoresis (PFGE) revealed a chromosome translocation between chrVIII and chrIX and an additional copy of chrIX in SparLox83 (Supplementary Fig. 2). In addition, we also identified 1197 SNPs/InDels in nuclear genome when we aligned the NGS reads to the yeast reference genome (GCF_000146045.2) (Supplementary Data 3). Excluding the sequences introduced by us, 305 SNPs/InDels were found to be associated with 158 overlapping open reading frames (ORFs). Notably, significant frameshift mutations were observed in *YAL063C*, *YBR005W*, *YBR148W*, and *YEL042W*.

The fitness of SparLox83 was evaluated by comparing growth in various conditions with its parental strain JDY524 (without loxPsym) and BY4741 in serial dilution tests. Growth of SparLox83 was indistinguishable from JDY524 or BY4741 on YPD and under stress conditions, including the presence of 8 µg/ml camptothecin (an inhibitor of topoisomerase I that causes genome instability) or 2.5 µg/ml nocodazole (a drug that affects mitosis by interference with microtubule polymerization) (Fig. 1b). Given the presence of multiple loxPsym sites in its genome, the genome stability of SparLox83 was checked using a previously described junction PCR-based assay[33]. None of the loxPsym sites were lost in 35 independent lineages after ~125 mitotic generations in YPD in the absence of Cre (Fig. 1c and Supplementary Fig. 3). These results confirmed the successful insertion of 83 loxPsym sites into the wild-type yeast genome without impairment of genome stability or fitness under various conditions.

### Induction of Cre recombinase activity in SparLox83R leads to versatile rearrangements, particularly inter-chromosomal rearrangements

To facilitate the selection of clones with genomic rearrangements, the previously described ReSCuES system[23] was integrated into SparLox83 to generate SparLox83R. ReSCuES was a reporter to efficiently identify SCRaMbLEd cells based on a loxP-mediated switch of "on" and "off" states of the two auxotrophic markers, *URA3* and *LEU2*. Since there is at

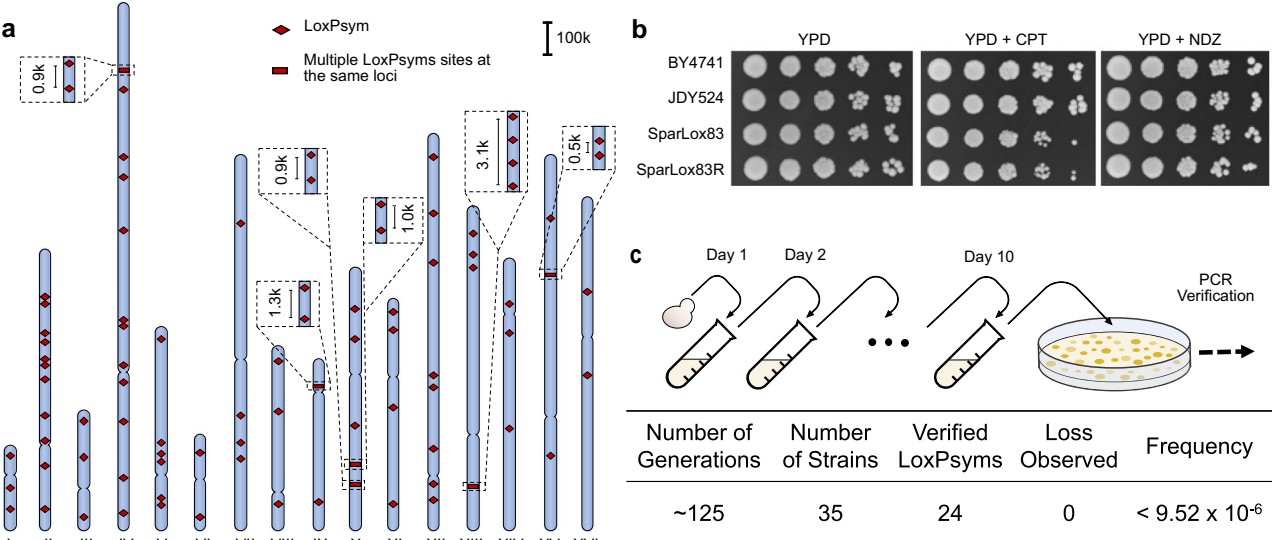

**Fig. 1 | Optimization, construction, and characterization of a yeast strain with loxPsym sites distributed throughout the genome. a** Distribution of 83 loxPsym sites in the SparLox83 strain. Detailed site locations are in Supplementary Data 1. The red diamond represents single loxPsym sites, while the red rectangle indicates multiple loxPsym sites located at the same loci as denoted within the dashed box. **b** SparLox83 phenotyping on various media. Ten-fold serial dilutions of overnight cultures of SparLox83R, SparLox83, parental strain JDY524, and wild-type BY4741 were analyzed on YPD, YPD + camptothecin (CPT, 8 µg/mL), and YPD + nocodazole (NDZ, 2.5 µg/mL). YPD, yeast extract peptone dextrose. Images taken after two days incubation at 30 °C. **c** PCR analysis of SparLox83 after ~125 generations to assess genomic stability. Potential loss of loxPsym was evaluated at 24 different loci in 35 passaged strains and no losses were observed. The frequency is a maximum estimate of loxPsym loss frequency per generation.

least one essential gene positioned between each pair of adjacent loxPsym sites, deletions will result in cell death, limiting the extent of all the possible rearrangements[34]. Therefore, an additional heterozygous diploid strain, JDY536, was generated by crossing SparLox83R with BY4742.

SparLox83R and JDY536 were induced to express Cre recombinase and selected using the ReSCuES system for five consecutive rounds before collection of SCRaMbLEants. No other selection was imposed. First, to confirm the presence of rearrangements, PCR was used to amplify some of the novel junctions that were not present in the original SparLox83R genome but which might be generated following loxPsym-derived rearrangements (Supplementary Fig. 4a, primers listed in Supplementary Data 4). Next, the pooled SCRaMbLEants were subjected to Nanopore MinION sequencing. To avoid PCR bias and obtain reads that were as long as possible, a ligation-sequencing strategy with no PCR amplification was used, and genome shredding steps were avoided. In total, 774,791 (8.06 Gb) and 2,523,754 (22.5 Gb) reads were generated from the SparLox83R and SparLox83R × BY4742 SCRaMbLEant populations, respectively.

A population-based strategy was adopted to detect and characterize rearrangement events in SCRaMbLEant populations (Fig. 2a). Flanking regions of loxPsym sites were extracted and assembled with all potential rearrangements in silico. Next, Nanopore reads containing loxPsym sequences were aligned to the potential rearrangements. Each filtered match between a read and a potential rearrangement was defined as a rearrangement event. This analysis revealed that 5721 and 936 rearrangement events took place across all loxPsym sites in the haploid and diploid strains, which accounted for 31.1% and 3.1% detected loxPsym-containing reads, respectively (Fig. 2b). Flanking region lengths were set to 500, 1000, 1500, and 2500 bp, with no significant differences seen in the analysis results (Supplementary Fig. 4b). The following results are those obtained using 1000 bp flank regions. At least one rearrangement event was detected for every loxPsym site in both the haploid and diploid strains. Interchromosomal rearrangements predominated: 4852 (84.8%) and 595 (63.6%) of the rearrangement events were inter-chromosomal in

haploid and diploid cells, respectively. Rearrangements were next assigned to their different chromosomes to test whether rearrangement events occurred evenly across the genome. In haploid cells, an increased number of rearrangement events was associated with a higher number of loxPsym sites on that chromosome (Fig. 2c, $R^2 = 0.9828$, $p < 0.001$).

These results obtained in the absence of selection beyond the requirement for at least one ReSCuES inversion event per cell showed that genome-wide insertion of loxPsym sites in SparLox83R produced versatile rearrangements, with a bias towards inter-chromosomal events, upon Cre induction. The preference for inter-chromosomal rearrangements suggests that SparLox83R presents an ideal model for studying the impacts of large-scale genomic rearrangements.

## Quantitative analysis revealed an uneven distribution of rearrangements at different loxPsym sites

Previous studies used rearrangement event frequencies at particular loxPsym sites to quantify their rearrangement capacities[31,32]. Rearrangement event frequencies were, however, biased by the different sequencing depths at each loxPsym site. To overcome this, an Average Rearrangement Rate (ARR), a normalized rearrangement event frequency, was used to quantify the average rearrangement potential for each loxPsym site. ARRs were compared for intra- and inter-chromosomal rearrangements. Although intra-chromosomal ARRs were significantly higher than inter-chromosomal ARRs, the difference between the two categories was limited (Fig. 2d). Moreover, outlier ARRs represented particularly active sites. Seven outliers (II-7, II-6, VIII-2, VIII-1, XII-8, VIII-3, XII-7) were found for intra-chromosomal ARRs in diploid cells, two of which (II-7 and II-6) were also observed in haploid cells. Three sites (VIII-2, XII-7, XIII-7) were identified as inter-chromosomal outliers in diploid cells, but no outliers were observed for inter-chromosomal rearrangements in haploid cells.

Rearrangement Weight (RW) was derived to quantitatively describe the rearrangement event frequency between two specific sites and reduce the bias caused by sequencing depth. RW divides the number of rearranged events by the geometric mean of the

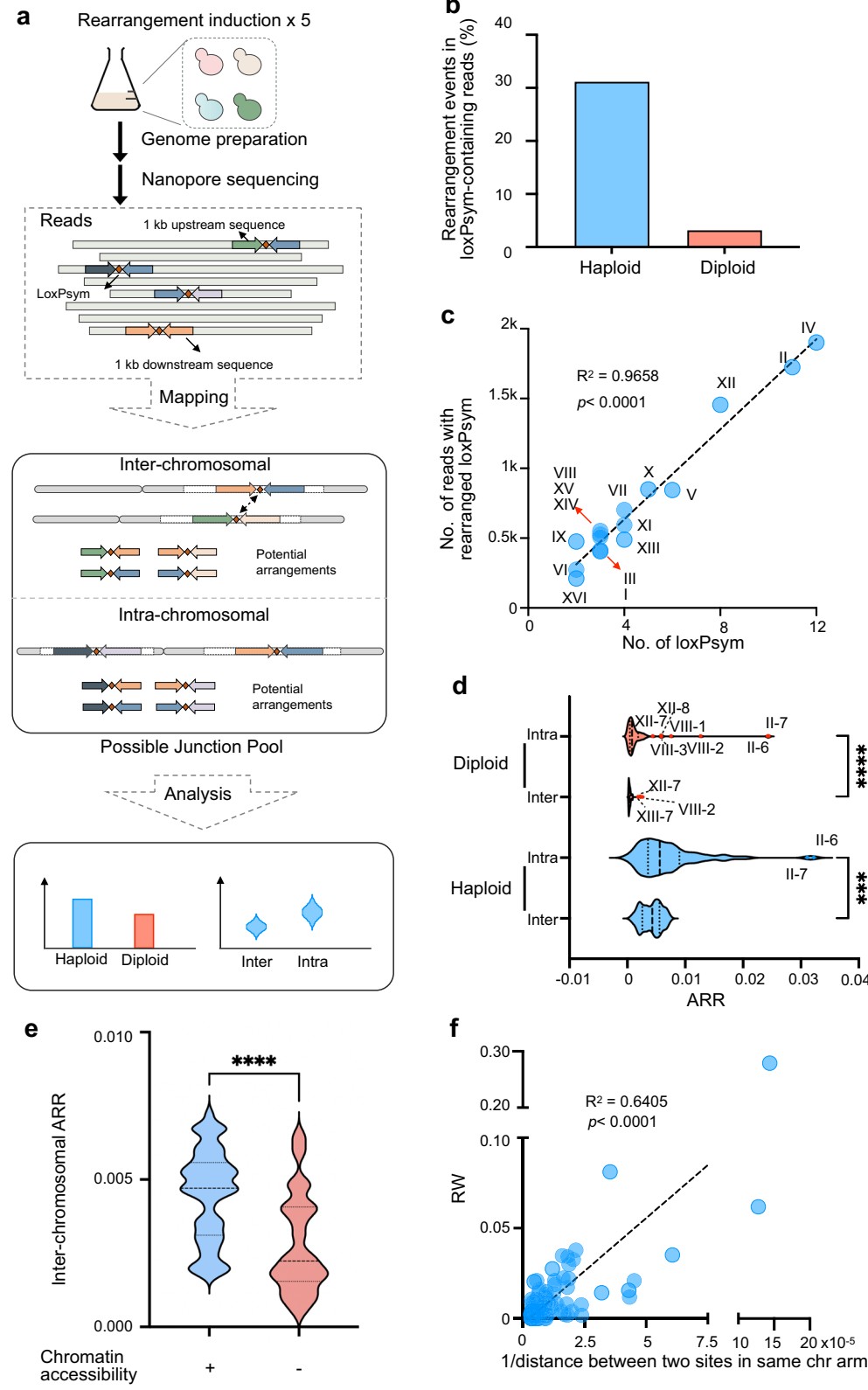

sequencing depths of pairs of interacting sites. The RW matrix containing RWs of all possible loxPsym site pairs (2775 pairs) was generated. Of these, 2111 (76.1%) non-zero RWs were represented in haploid cells, suggesting that most of the potential rearrangement events occurred. Interaction maps of the RW matrix for intra- and interchromosomal rearrangements were generated (Supplementary Fig. 5). The RW between sites II-6 and II-7 (0.28) was significantly higher than

other RW values, which were below 0.087 (Supplementary Fig. 5a). Inter-chromosomal RWs also exhibited an uneven distribution (Supplementary Fig. 5b), with events between site pairs located in the inner circle (RWs > 0.02) occurring more frequently than with other site pairs (RWs < 0.02). Together, these results showed that loxPsym sites in SparLox83R exhibited highly variable rearrangement event frequencies.

**Fig. 2 | Characterization of genome-wide rearrangement by SCRaMbLE.**
**a** Workflow for loxPsym-mediated genome-wide rearrangement and analysis. A single colony was cultured and subjected to five sequential rounds of Cre recombinase induction to generate a SCRaMbLEant pool. The genomic DNA was then sequenced. Potential rearrangements were identified and analyzed by mapping the loxPsym-containing reads to the sequence pool comprising all possible pairwise combination sequences 1 kb upstream (forward arrow) and 1 kb downstream (backward arrow) of each loxPsym site. **b** Percentage of reads exhibiting genomic rearrangements in detected loxPsym-containing reads in haploid (blue) and diploid (red) SCRaMbLEant cells. **c** Correlation between the number of identified rearrangements and the number of loxPsym sites in each chromosome. Y-axis, number

of reads with rearrangement; X-axis, number of loxPsym sites in a chromosome. Simple linear regression was used. $R^2 = 0.9658$, $p < 0.0001$. **d** Average rearrangement rate (ARR) in haploid (blue) and diploid (red) cells. Mann–Whitney test was used to compare ARRs between intra- and inter-rearrangement. ***$p = 0.0003$; ****$p < 0.0001$. **e** Comparison of inter-chromosomal ARRs between loxPsym sites within (+, blue) and outside (−, red) open chromatin regions in haploid cells. Unpaired t-test (two-tail) was used to compare the two groups, ****$p < 0.0001$. **f** Correlation between Rearrangement Weight (RW) and 1/genomic distance of loxPsym sites in the same chromosome arm in haploid cells. Simple linear regression was used. $R^2 = 0.6405$ and $p < 0.0001$. Source data are provided as a Source data file.

## The genomic locations of loxPsym sites impact their rearrangement event frequencies

Previous studies using SCRaMbLE in strains harboring one or more synthetic chromosomes have confirmed that rearrangement frequency varied among loxPsym sites[31,32]. In synthetic chromosomes, where loxPsym sites are densely arranged, the rearrangement frequencies of loxPsym sites correlated with chromatin accessibility and spatial contact probability[32]. However, due to the preference for intrachromosomal rearrangements in these synthetic chromosomes, factors impacting inter-chromosomal rearrangement frequencies remain less well understood. The genome-wide distribution of loxPsym sites in SparLox83R, with their frequent inter-chromosomal rearrangements and quantitatively defined RWs, indicate that SparLox83R is an ideal model to explore the features of rearrangement frequencies involved in sparsely arranged loxPsym sites.

Chromatin accessibility refers to the degree physical accessibility and availability of DNA within the chromatin structure for interaction with various regulatory factors, such as transcription factors and other proteins. It describes the ability of these factors to bind specific genomic regions and regulate gene expression[35]. Thus, it may be a factor to influence the interaction between the Cre recombinases and loxPsym sites. Consistent with previous findings[32], the inter-chromosomal ARRs of sites positioned at open chromatin regions[36] were significantly higher than sites outside open chromatin regions (Fig. 2e), while no significant correlation was observed for intra-chromosomal ARRs (Supplementary Fig. 6).

The highest RW was observed between sites II-6 and II-7, which possessed the shortest genomic distance among the 83 loxPsym sites. Next, the impact of genomic distance between two intra-chromosomal loxPsym sites on RW was examined. The RW between two loxPsym sites located on the same chromosome arm showed a negative correlation with genomic distance in haploid and diploid cells (Fig. 2f and Supplementary Fig. 7a). For two sites not on the same chromosome, spatial distance was determined using genome-wide chromosome conformation capture analysis (Hi-C), which facilitated the generation of a normalized contact map of the SparLox83R genome (Supplementary Fig. 7b). However, no correlation was found between the normalized contact counts and the RW of paired loxPsym sites (Supplementary Fig. 7c).

Together, these results suggest that chromatin accessibility and genomic distance partly impact the rearrangement frequency of loxPsym sites.

## Large-scale genomic rearrangement and stress tolerance

Large-scale rearrangements drive phenotypic diversification and environmental adaptability[37,38], but distinguishing the impact of large-scale rearrangements on fitness from other types of mutation in naturally evolved strains remains challenging. The SparLox83R strain can be used to introduce large-scale genomic rearrangements, allowing the investigation of the role of such rearrangements in cell fitness alterations under various conditions.

Rearrangements were induced in SparLox83R and strains were screened under a range of selective stresses including high alcohol

(10% ethanol), high oxidation (3 mM $H_2O_2$), high osmotic stress (1 M NaCl), and non-fermentable sugar as a carbon source (3% glycerol). Experiments were repeated in the presence of reagents to perturb microtubules (10 µg/ml nocodazole and 40 µg/ml benomyl), cause DNA damage (30 µg/ml camptothecin), or inhibit the TOR pathway (10 ng/ml rapamycin) (Fig. 3a)[23]. Although no candidates conferring increased tolerance to ethanol, $H_2O_2$, NaCl, or glycerol were identified (Supplementary Fig. 8a), several clones were more resilient to nocodazole, benomyl, or rapamycin than the original strain (Fig. 3b and Supplementary Fig. 8b).

One strain, JDY528, which gained tolerance to nocodazole, was selected for further analysis. Whole-genome sequencing was conducted using the Oxford Nanopore MinION platform. Genome assembly using Canu and correction using Nanopolish enabled comparison with the reference genome and identification of two loxPsym-mediated genome arrangements in the JDY528 strain. A translocation was observed between chromosomes IV and XIV (mediated by IV-5 and XIV-1), and a 484 kb duplication was found on chromosome IV (mediated by IV-7 and IV-9) (Fig. 3c). In addition, the additional copy of ChrIX in SparLox83R was lost in JDY528 (Supplementary Fig. 9a). The presence of these chromosomal rearrangements was confirmed by junction PCR (Supplementary Fig. 9b).

Major causal mutations in response to such selective forces are expected to change gene location/context and may involve the formation of new chromosomal junctions that affect higher-order genome regulation[39]. Therefore, we investigated the three-dimensional (3D) genome structure and transcription profile in these strains by Hi-C and RNA-seq analysis.

To determine the impact of rearrangements on spatial location, genome-wide Hi-C contact maps[40] were compared between JDY528 and SparLox83R. Clustering of centromeres and telomeres (Supplementary Fig. 9c, red dots on the map) was seen in both strains, as previously reported[41,42]. Although one of the rearranged loxPsym sequences in JDY528 was near the centromere of chrIV, the contact maps showed that the translocation and duplication events in JDY528 did not disrupt localization and clustering of centromeres and telomeres (Supplementary Fig. 9d). The $log_2$-ratio heatmap between JDY528 and SparLox83R was plotted[43] to quantitatively compare the differences between the two strains. This analysis showed that contacts between chromosomes without genomic rearrangement in JDY528 and SparLox83R were similar (Fig. 3d). However, the duplication on chromosome IV led to elevated contacts with the duplicated region. The translocation between chromosome XIV and chromosome IV altered the contacts between these two chromosomes (Fig. 3d, black circle). Moreover, the contacts between chromosome IX and other chromosomes were weaker in JDY528 than in SparLox83R, which can be attributed to the loss of one copy in JDY528 (Fig. 3d, black arrow).

To investigate the genome-wide transcriptional impact of nocodazole treatment on cells cultured in YPD media, RNA sequencing was performed due to the tight relationship between chromosome

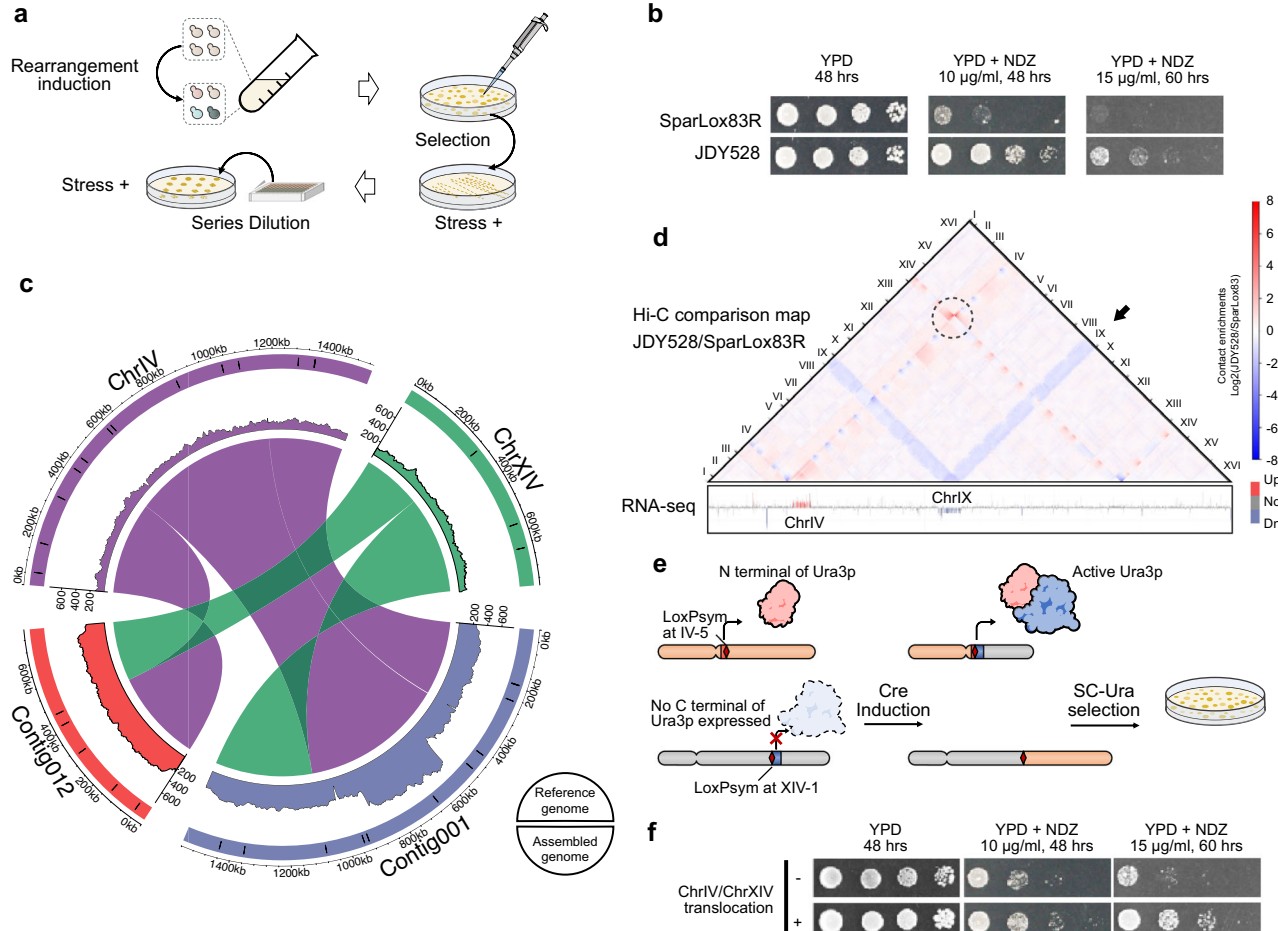

**Fig. 3 | Genome-wide rearrangements confer resistance to various stresses.**
**a** Workflow used to identify strains tolerant to various stresses. A single colony was cultured and subjected to rearrangement induction before being plated onto media containing drug, alcohol, oxidation, or osmotic stress treatments. Colony phenotypes were verified using serial dilution assays. **b** Ten-fold serial dilutions of overnight cultures of the original strain (SparLox83R) and a nocodazole-resistant clone (JDY528) were analyzed on YPD and YPD containing different concentrations of nocodazole (YPD + NDZ). NDZ, nocodazole. **c** Colinearity mapping between assembled contig001 (blue) and contig012 (red) in JDY528 with chrIV (purple) and chrXIV (green) of the reference genome. The outer circle represents chromosomes before (upper half) and after (lower half) translocation. Black bars in the outer circle indicate the locations of loxPsym sites. The middle circle is a depth plot of the

Nanopore sequencing data, and the inner circle indicates the inferred structural variants. **d** Ratio plots of contact maps and RNA-seq analysis of the SparLox83R and JDY528 genomes. For the Hi-C data, two independent biological repeats were performed and analyzed. A translocation (dotted circle) and chromosome position changes (black arrow) were indicated. The color code in heatmap denotes contact enrichments. For the RNA-seq data, three independent biological repeats were performed and analyzed. Red indicates increased abundance and blue indicates decreased abundance. **e** Representation of the split-*URA3* method used to reconstruct the chrIV/chrXIV translocation in JDY524 (i.e., the parental strain lacking loxPsym sites). **f** Ten-fold serial dilutions of strains with (+) or without (−) the chrIV/chrXIV translocation analyzed on YPD and YPD with different concentrations of nocodazole (YPD + NDZ).

organization and gene expression. A total of 197 gene transcripts with significantly altered expression levels (>2-fold increase or decrease, *p* value < 0.0001) were identified in JDY528. Among the 82 upregulation genes, 73 genes were located within the duplication region (from *YDR073W* to *YDR277C*), while among the 115 downregulation genes, 75 genes were found on chromosome IX (Supplementary Fig. 10a). To determine whether these differentially expressed genes were associate with the large fragment rearrangement, we combined the RNA-seq data and Hi-C contact map, revealing that most of these genes clustered around either the rearrangement sites or chromosome IX (Fig. 3d). These findings strongly suggested that alterations in the 3D genome structure may contribute to changes in gene expression. Furthermore, we observed a decreased expression level of *YDR001C* which was located near the translocated junction (Supplementary Fig. 10a). However, its knock-out strain exhibited comparable tolerance to nocodazole when compared to wild-type strain (Supplementary Fig. 10b). Considering that changes in expression levels were not limited to regions

surrounding translocation breakpoints, further analyses should explore a larger set of potentially involved genes. In addition, it was possible that nocodazole resistance could be influenced by both the large-scale duplication on ChrIV and the copy number variation of ChrIX.

To test whether the variation in chromosomal structure was sufficient to cause the observed nocodazole tolerance, a split-*URA3* reporter strategy[25] was used to reconstruct the chrIV/chrXIV translocation of JDY528 in JDY524 (the initial strain lacking loxPsym sites, Fig. 3e). The strain carrying only this translocation grew better than the split-*URA3* inserted into JDY524 on medium with nocodazole (Fig. 3f), providing direct evidence that the chrIV/chrXIV translocation contributed to the observed nocodazole tolerance. These data revealed a connection among large-scale rearrangement, genome 3D structure, transcription, and phenotype, demonstrating that simply reconfiguring chromosome architecture was sufficient to provide fitness advantages in stressful growth conditions.

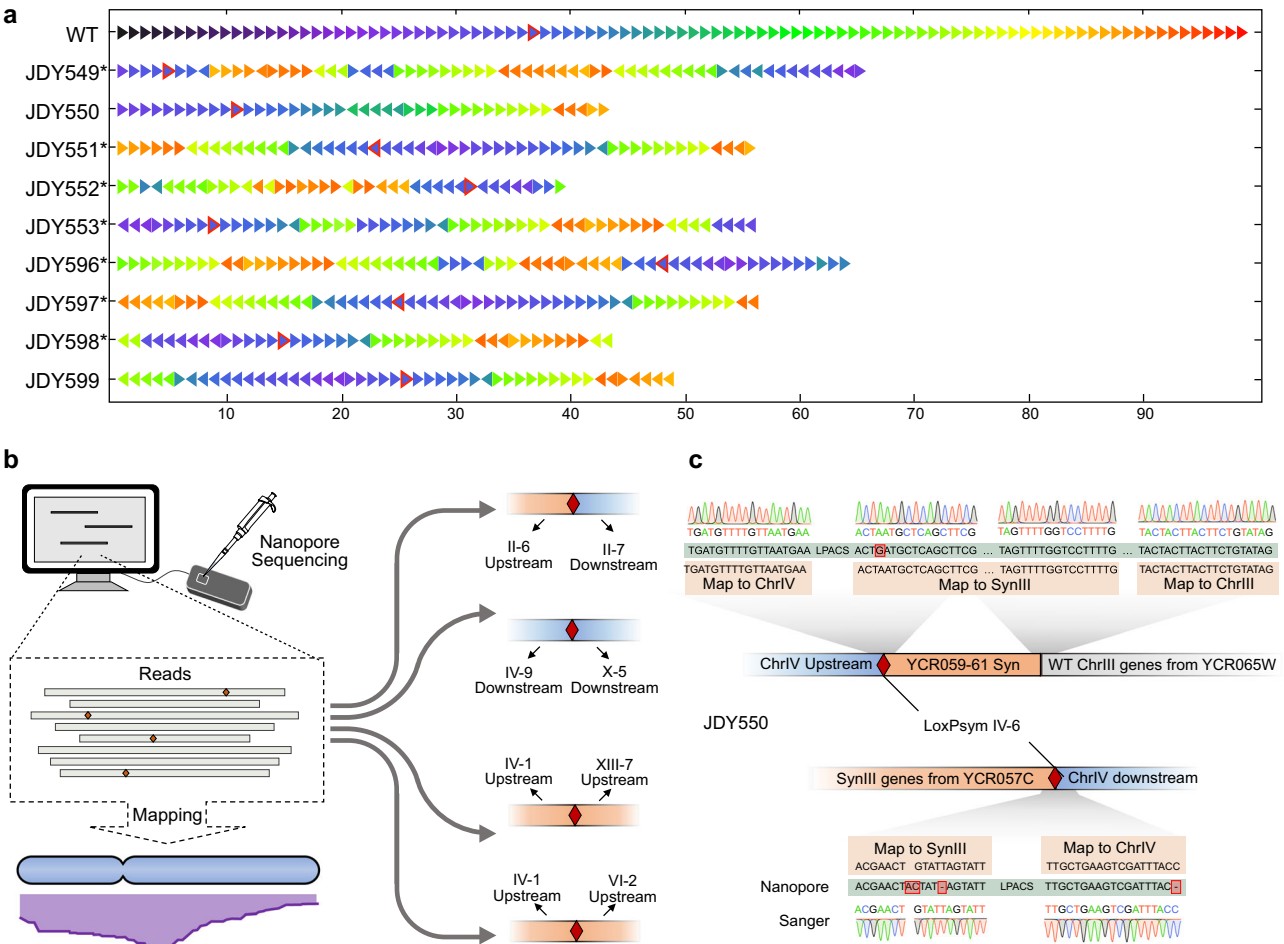

**Fig. 4 | Structure variations caused by whole-genome-wide SCRaMbLE in diploid strains. a** SynIII rearrangements detected in diploids. Arrows indicate segments between two adjacent loxPsym sites. Arrow colors in the SCRaMbLEgram indicate the segment number in the parental chromosome, and arrow directions indicate orientation. Arrows with red borders denote segments containing centromeres. Strains with an asterisk (*) have a circular synIII as a result of SCRaMbLE. **b** Schematic of Nanopore sequencing analysis of chromosome changes. A deletion between II-6 and II-7 was found in JDY550. A translocation between chrIV and chrX mediated by IV-9 and X-5 was found in JDY549 and JDY598. A translocation between chrIV and chrXIII mediated by IV-1 and XIII-7 was found in JDY551, JDY597, and JDY599. A translocation between chrIV and chrVI mediated by IV-1 and VI-2 was found in JDY552. The upstream region of a loxPsym site is indicated by the color red, while the downstream region is indicated by the color blue. **c** A complex rearrangement event among chromosomes III, IV, and synIII in JDY550. The translocation occurred at the IV-6 site and involved synIII and chrIII. This rearrangement was confirmed by both Nanopore and Sanger sequencing. Green bars show the sequence of one typical Nanopore read. LPACS, loxPsym, and adjacent co-transformed sequence.

## Chromosomes harboring sparsely arranged loxPsym sites combined with synthetic yeast chromosomes can generate complicated structural variations

SCRaMbLE is widely used in directed evolution to acquire strains with distinctive traits[21–23,25,27,34]. Here, to boost SCRaMbLE with genome-wide large-scale rearrangements, SparLox83R chromosomes were combined with a Sc2.0 synthetic chromosome in a single strain. As a proof of concept, SparLox83R was mated with JDY541, the strain carrying synIII[33], to generate JDY544. After five rounds of SCRaMbLE using ReSCuES without selective pressures, nine SCRaMbLEant colonies were randomly isolated and sequenced. The presence of loxPsym sites and PCRTags in synIII and chrIII allowed reads with loxPsym sites to be extracted and differentiated in SparLox83R and synIII.

First, SCRaMbLEd synIII was resolved in the sequenced strains. The arrangement of synIII in each strain was deconvoluted and visualized as described previously[18] and each strain exhibited a unique structure (Fig. 4a). Rearrangement types in the nine strains were classified as deletions, inversions, duplications, duplicated inversions, inverted duplications, and complex rearrangements with respect to the parental synIII sequence. As an example, the pattern of

rearrangements detected in JDY553 and JDY596 were illustrated as a dot-plot, wherein the segments of the rearranged genome were plotted against the corresponding segments of the parental genome in their original sequential order (Supplementary Fig. 11a). Meanwhile, the fate of each segment in each strain was determined (Supplementary Fig. 11b and Supplementary Table 1). By comparing sequence order and orientation in all strains except JDY550, several commonly conserved sequences (orange) and non-conserved sequences (gray) were identified (Supplementary Fig. 11c), most of which were consistent with previously reported regions[30]. The regions at the ends of synIII were prone to delete, but to different degrees in different strains (Supplementary Fig. 11c, light gray rectangles). Seven of the nine strains harbored a circular synIII, which was also observed in SCRaMbLEant cells previously[44] (Fig. 4a, see asterisks). This circularization was confirmed by reads spanning the left and right ends of synIII. In addition, whole-chromosome duplications were identified in three strains (JDY550, JDY596, and JDY599).

For chromosomes other than chromosome III and synIII, reads containing loxPsym sites were aligned to the reference sequences of SparLox83R. Four types of structural variations were observed in seven

of the nine strains, including one deletion event and six translocation events (Fig. 4b). In addition, a structural change was found in JDY550 mediated by IV-6 and loxPsym site 73, located between *YCR057C* and *YCR059C* on synIII. Read alignments revealed that IV-6-Up was combined with segment 74 on synIII, and IV-6-Down combined with segment 73 (Fig. 4c). A rearrangement between synIII and chrIII, in which the *YCR061W* gene was from synIII and its neighboring gene *YCR065W* was from chrIII (Fig. 4c), was also observed. However, no loxPsym sequence were detected at the breakpoint, which suggested that the rearrangement was generated by homologous recombination as synIII and chrIII shared significant sequence similarity. Three primer pairs were used to confirm these structure changes and the expected amplicons were directly sequenced to verify the specific translocation sites (Fig. 4c and Supplementary Fig. 12). These results demonstrated that loxPsym sites in SparLox83R were able to combine with synthetic chromosomes to generate a range of structural variants genome wide, including rearrangements between SparLox83R-derived loxPsym sites and synIII-derived loxPsym sites and one example of an homologous recombination-mediated rearrangement (Table S1).

### LOH and aneuploidy are frequently detected in SCRaMbLEd heterozygous diploids

As the rearrangement frequency was relatively lower with SparLox83R-derived loxPsym sites than with synIII-based sites, it was doubtful whether loss of heterozygosity (LOH) and aneuploidy, which were observed in previous SCRaMbLEd genomes[20], were occurring in these strains. To examine this, the 2 kb upstream or downstream regions of each loxPsym site were designated as loxPsym site-Up and loxPsym site-Down and were analyzed as a unit. For example, X-1-Up and X-1-Down indicated the 2 kb upstream and downstream regions of X-1. Reads with or without loxPsym sites were mapped to the upstream/downstream regions and counted to deduce copy number variation of regions adjacent to loxPsym sites (Fig. 5a; where a blue square represents a copy of the upstream region or downstream region containing a loxPsym site and a white square represents a region lacking a loxPsym site). Using this strategy, copy numbers for all loxPsym upstream and downstream regions were assessed and some LOH events were observed in all nine strains (Fig. 5b and Supplementary Fig. 13). Several LOH events were likely caused by loxPsym site sequence deletion, such as the loxPsym sites on chromosome II (except for II-4 in JDY553 and JDY598), whereas other events were likely caused by loxPsym site sequence insertions into the counterpart of the other chromosome, such as IV-1 in all strains (except JDY552) (Fig. 5b). In addition, the structure of chrVIII in all nine strains differed from that of the parent strain JDY544. In JDY549, JDY550, and JDY 596, only one copy of the wild-type chrVIII remained (Fig. 5b, c). In JDY551, JDY597, JDY598, and JDY 599, two copies of the wild-type chrVIII were present instead of the heterozygous chrVIII in JDY544 (Fig. 5b, c). In JDY552, the loxPsym sequence of VIII-1 was inserted into the wild-type chrVIII and the remainder of SparLox83R-based chrVIII was lost (Fig. 5b, c). JDY553 contained three copies of the wild-type chrVIII, one of which contained an insertion of the VIII-1 loxPsym sequence (Fig. 5b, c). These variations were likely caused by structural differences in chrVIII between SparLox83R and the wild-type, in which a translocation of VIII-3 and IX-1 pre-existed prior to SCRaMbLEing (Supplementary Fig. 2).

Using the copy number variation information, the chromosomal structural variations detected by WGS were reconstructed and several non-reciprocal translocations were identified. In JDY549 and JDY596, an extra copy of IV-9-Down combined with X-5-Down and formed a new chromosome, with deletion of X-5-Up (Figs. 4b and 5d). In JDY551, JDY597, and JDY599, an extra copy of IV-1-Up combined with XIII-7-Up, forming a new chromosome and deleting XIII-7-Down (Figs. 4b and 5d). In JDY552, however, a duplication of VI-2-Up was observed with no change to IV-1-Up (Fig. 5b). To clarify the events at these sites, reads were aligned to reference chromosomes. A translocation occurred

between loxPsym sites IV-1 and VI-2, and a duplication of an ~45-kb region upstream of VI-2 was inserted upstream of IV-1. In this case, it was uncertain whether the rearrangement was loxPsym-mediated (Fig. 5e).

Overall, the diverse rearrangements and resulting genomes indicated that sparsely arranged loxPsym sites in SparLox83R can be used to generate combinatorial diversity with synthetic chromosomes, including structural variation, LOH, and aneuploidy.

### SparLox83R accelerates SCRaMbLE-mediated enhancement of acetic acid tolerance

The increased rearrangements observed in strains with synIII, as exemplified by the diversity of outcomes generated by SCRaMbLE of SparLox83R and synthetic chromosomes, suggested a potential strategy for strain improvement under stress conditions. To test this, the rate of variant evolution was compared between two heterozygous diploids, JDY546 (BY4741 × synIII) and JDY544 (SparLox83R × synIII), using tolerance to acetic acid (HAc) as a proof of principle. In total, 48 individual colonies from each strain were inoculated into a 96-well microplate and SCRaMbLE-Selection-based evolution cycles were performed. Mixed strains after each SCRaMbLE round were selected on YPD plates with a HAc gradient (Fig. 6a).

The original strains (SparLox83R, JDY544, and JDY546) were tested for HAc tolerance to identify an appropriate level of stress. Severe growth defects were seen in the presence of 0.4% HAc (Supplementary Fig. 14a), indicating that this concentration was appropriate for selection of tolerant SCRaMbLEants. After the first round of SCRaMbLE (R1), most of the populations were able to grow on media with a maximum of 0.4% HAc (Supplementary Fig. 14b). After the second (R2) and third (R3) rounds, the HAc-tolerance of all populations increased significantly (Supplementary Fig. 14c, d). After the second round, most SCRaMbLEant populations were able to grow on 0.5% HAc media, with populations from JDY544 exhibiting better growth than those from JDY546 (Supplementary Fig. 14c). After the third round (R3), two different concentrations of cells were tested on YPD plates with a gradient of HAc ranging from 0.5% to 0.8% (Fig. 6b and Supplementary Fig. 14d). The SCRaMbLEant populations from 48 individual JDY544 colonies were able to survive on 0.7% HAc, whereas only some of the 48 individual JDY546 colonies could grow under the same conditions. These results suggested that the development of increased HAc tolerance was greatly accelerated in SCRaMbLEant populations derived from JDY544 compared with those from JDY546, indicating that genomes with sparsely distributed loxPsym sites may be associated with rapid evolution of tolerance to HAc.

## Discussion

In this study, a strain containing 83 loxPsym sites sparsely arranged throughout all chromosome arms was constructed and used to study genomic rearrangement. Pooled sequencing of haploid and diploid SCRaMbLEants revealed that comparable inter- and intra-chromosomal rearrangement events were detected after inducing Cre recombinase expression. Our results indicated that genome ploidy and genomic location of loxPsym sites were the major factors that impacted rearrangement ability. Further examination revealed that one of the inter-chromosomal translocations altered the 3D genome structure and was associated with increased tolerance to environmental stress. When combined with SCRaMbLE in diploids made by crossing SparLox83R and a synIII-containing strain, LOH and aneuploidy, as well as structure variations, frequently occurred in the heterozygous diploid cells. The SparLox83R-derived chromosomes in the diploids exhibited increased genome instability after Cre recombinase induction and generated diverse rearrangements to facilitate more rapid environmental adaptation than diploids only with wild-type chromosomes.

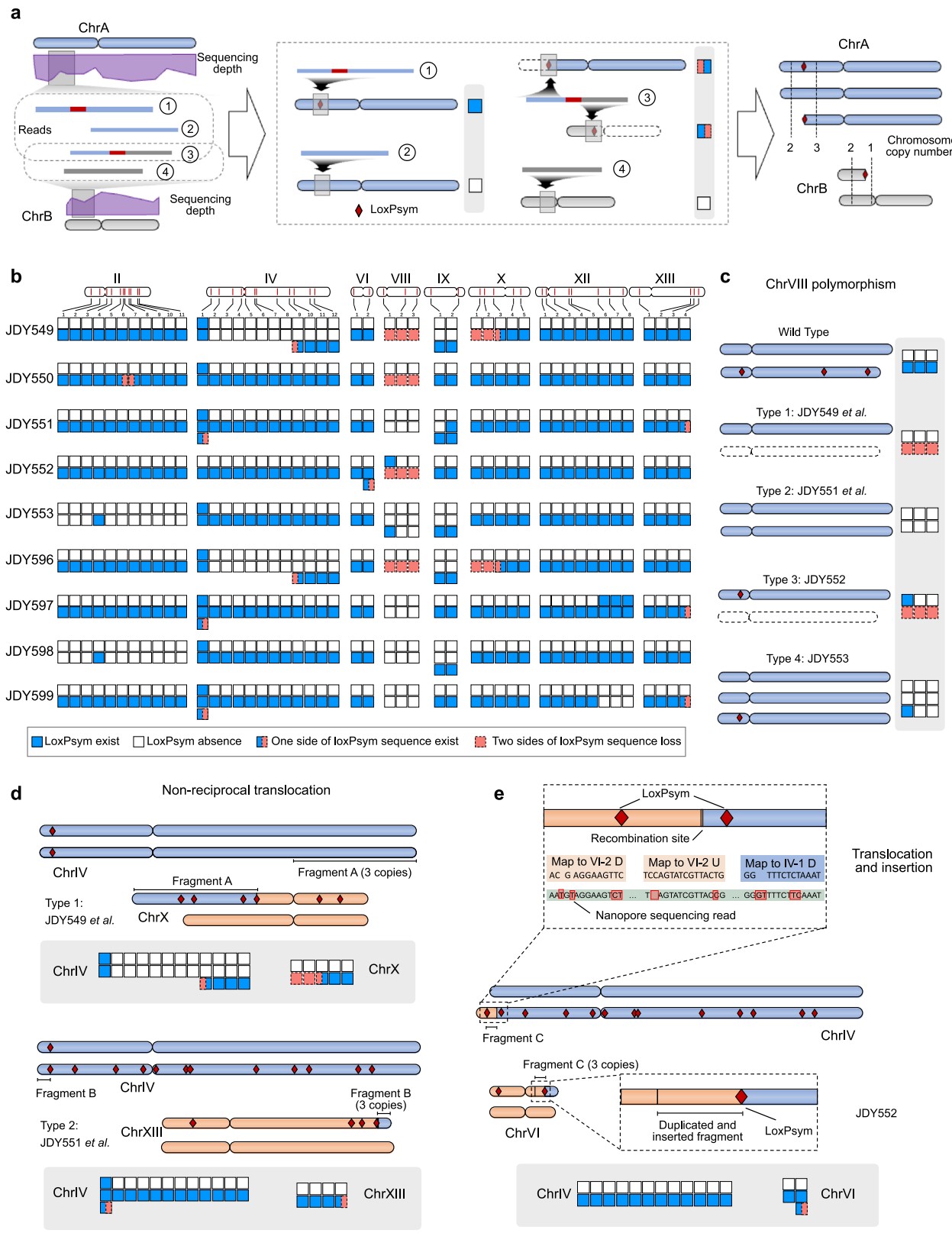

Genomic rearrangement is a major driver for rapid chromosomal evolution and the utility of the Cre-loxPsym system provides an artificial strategy to increase the opportunities for rearrangement[16]. Previous studies have shown that several factors are associated with the recombination frequency mediated by the Cre-loxPsym system, including genome ploidy, chromosomal location of loxPsym sites, the spatial proximity of interactive loxPsym sites, and 3D chromosome

conformation[18,29,31,32]. Sequencing of pooled cells showed that loxPsym sites in SparLox83R exhibited uneven rearrangement event distribution and multiple rearrangement outliers existed in both haploid and diploid strains. The intra-chromosomal outliers, mainly via the II-6 and II-7 loxPsym sites, were caused by higher rearrangement preference between these sites, which were only 7 kb apart and were therefore potentially situated in the same chromosomal interacting domain

**Fig. 5 | LOH and aneuploidy caused by genome-wide SCRaMbLE in diploids.**
**a** Workflow of copy number variation analysis. Reads with or without loxPsym sites were mapped to the upstream/downstream regions of each loxPsym site. The copy number of the upstream/downstream regions for a certain loxPsym site were deduced based on read alignment. Reads from corresponding chromosomes are indicated by the same color (blue or gray). Different rectangle colors indicate four distinct situations deduced from read mapping: blue indicates loxPsym is present and not rearranged, white indicates absence of loxPsym, orange with dash lines indicates loss of loxPsym-containing sequence and half blue with half orange squares indicate rearrangement. Structural variation was deduced using copy number calculation of loxPsym flanking sequences. **b** Copy number variation of 2 kb upstream/downstream regions of the loxPsym sequence in the nine diploid strains. Chromosomes lacking rearrangements were shown in Supplementary Fig. 13. **c** Three different types of chrVIII polymorphism detected in the nine strains. **d** Two types of non-reciprocal translocation observed between loxPsym sites IV-9 and X-5 or IV-1 and XIII-7. **e** A translocation occurred between two loxPsym sites, IV-1 and VI-2, and a duplication of the region upstream of VI-2 was inserted at loxPsym site IV-1. Reads revealed that the upstream sequences (U) of VI-2 were connected to the downstream sequences (D) of IV-1.

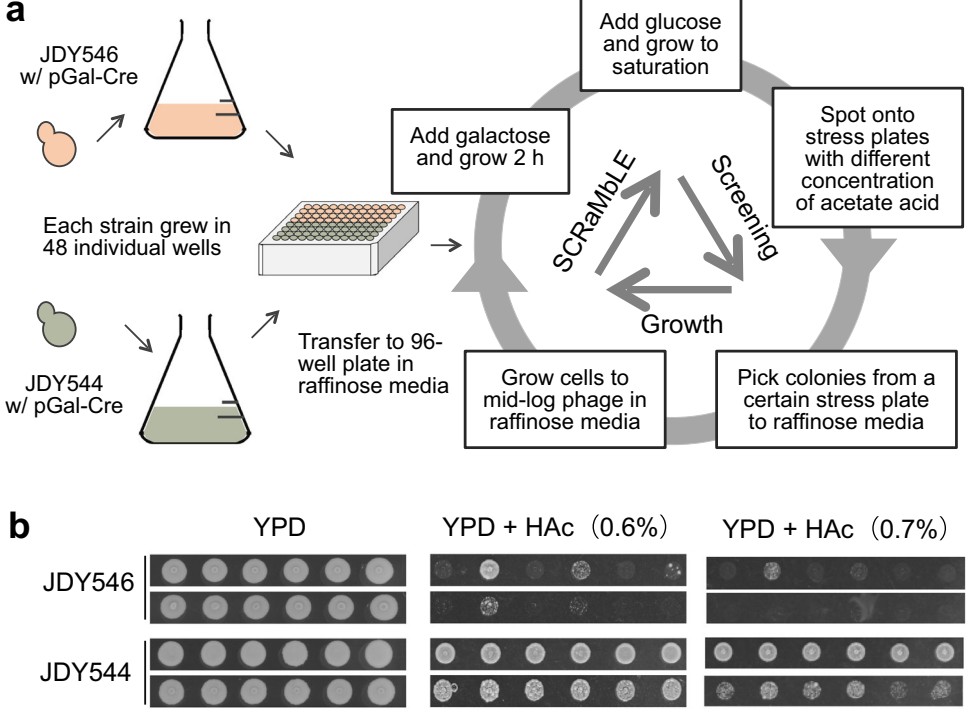

**Fig. 6 | Rapid improvement of acetic acid tolerance using genome-wide SCRaMbLE. a** Workflow of SCRaMbLE-Selection-based evolution cycles. Two het-erozygous diploids, BY4741 × synIII (JDY546, red) and SparLox83R × synIII (JDY544, green), were inoculated into 48 individual wells and cycle growth-SCRaMbLE-screening was applied to SCRaMbLEant populations. Populations were screened after each round of SCRaMbLE by selection on YPD with an acetic acid (HAc) gradient. **b** After three rounds of evolution, SCRaMbLEant populations from SparLox83R × synIII were more resistant to HAc than those from BY4741 × synIII.

(CID)[45]. Supporting this notion, a previous study reported that loxP-sym sites in the same CID actively rearrange with each other[29]. The mechanism of the Cre/loxP system relies on Cre recombinase func-tioning as a tetramer with two loxPsym sites, and our results are therefore consistent with the notion that the intra-chromosomal rearrangement frequency between two sites is correlated to their proximity[32]. However, no correlation was detected between proximity and inter-chromosomal rearrangement frequency[46]. Inter-chromosomal rearrangement may form dicentric chromosomes which can result in cell growth defect because dicentric chromosomes are instability at mitosis. Thus, the factors correlated to inter-chromosomal rearrangement are more complex than those of intra-chromosomal rearrangement. In addition, the distribution of cells with different rearrangements in the final population for sequence can be influenced by growth advantages or disadvantages resulting from structural variations. This factor also contributes to rearrangement preferences when utilizing the SCRaMbLEd pool for sequencing, although it is not easily distinguishable on its own.

Characterization of genetic rearrangements in SparLox83R and SparLox83R × BY4742 revealed that the proportion of rearrangement events was 10-fold higher in haploid than in diploid cells. There are

several possible reasons behind this phenomenon. First, the loss of loxPsym sites could only be occurred in diploid strains. In haploid strains, loxPsym-mediated deletion of fragments might lead to leth-ality because of essential gene loss[47,48] while deletions can occur in the diploid without lethal effects because of the additional presence of the wild-type genome[34]. In addition, the possibility of the occurrence of LOH and aneuploidy events have been demonstrated by sequencing nine SCRaMbLEant colonies (Fig. 5). This might also be happened in the diploid strain of SparLox83R × BY4742 and further decrease the total number of loxPsym sites[49]. Moreover, five rounds of induction-selection were performed before sequencing, and this may have fur-ther lowered the total number of loxPsym sites. Second, the additional copy of chromosome without loxPsym site may hinder the rearran-gement capacity of the inter-chromosomal interaction. Although diploid colonies based on synthetic chromosomes of Sc2.0 exhibited a higher recombinational frequency compared to haploid colonies, no inter-chromosomal events were detected among the hundreds of sequenced diploid colonies[31]. We also detected a related lower inter-chromosomal interaction proportion in diploid strains (63.6%) than in haploid strains (84.8%) while the expected proportion is 7% for intra-chromosomal events (210/2775) and 93% for inter-chromosomal

events (2565/2775) if there is no preference for intra- or inter-chromosomal recombination. In addition, haploid strains may have high tolerance to gross inter-chromosomal events as its plasticity. However, in a diploid strain, recombination events between different chromosomes may give rise to two distinct genotypes of the homologous chromosome pairs, possibly causing genomic instability and severe growth detect[50]. More experimental validations are needed.

Our introduction of SCRaMbLE in SparLox83R × synIII is the first trial to induce large-scale genomic rearrangement as well as SCRaMbLE in a diploid strain. Consistent with previous research[18], the arrangements of synIII in nine diploid SCRaMbLEants obtained using ReSCuES showed unique structures that differed from one another (Fig. 4a), demonstrating the capacity of SCRaMbLE. Moreover, several synIII arrangement traits were found in these strains. First, the remaining segments showed high similarity in all strains (Supplementary Fig. 7b), partly overlapping earlier research[30]. Segments 65, 66, and 67 were deleted in our strains but were retained in the haploid strain possessing an essential gene plasmid previous reported[30], indicating that synthetic lethal interactions can be bypassed when a copy of wild-type chrIII is present. Second, all diploid SCRaMbLEants harbored a shorter synIII, whereas about one-third of the haploid strains in previous studies harbored a rearranged synIII larger than the parental chromosome[18,19]. These genotypic differences could be due to our use of linear chromosomes in a diploid strain, by contrast with previous studies that used ring chromosomes in haploid strains[18,19]. These results indicate that, as previously noted[33], deletion is more likely in diploid strains and large fragment deletions can cause severe growth defects in haploid strains. In addition, we found that SCRaMbLE generated a high frequency of chromosome circularization, with numerous segments near to the telomere region deleted but with gene arrangements near the centromere region retained.

Only one rearrangement between the SparLox83R and synIII chromosomes was observed in the nine diploid strains (Fig. 4c). However, a higher rate of intra-structural variations within synIII was identified (Fig. 4a). This could be caused by several potential mechanisms. First, synIII has a higher density of loxPsym sites and the sites are closer to one another than sites on the SparLox83R chromosomes, likely resulting in increased interactions between Cre recombinases binding at neighboring sites. Second, SCRaMbLEant sequencing analysis revealed that intra-chromosomal ARR is significantly higher than inter-chromosomal ARR (Fig. 2d), and similar outcomes were also previously observed in strains harboring more than one synthetic chromosome[19] and a study based on Ty element mediated rearrangement[51]. Third, high efficiency of recombination in yeast may inhibit the rearrangement efficiency mediated by loxPsym sites in diploid cells because donors with long homologous arms are provided by wild-type chromosomes and many LOH events were identified (Fig. 5). Although limited interactions were observed between SparLox83R-derived chromosomes and synthetic chromosomes, strains harboring two genotypes of chromosomes exhibited an accelerated acquisition of HAc tolerance (Fig. 6). These findings suggested that exposure to challenging environments was crucial for identifying cells with genomic rearrangements when utilizing the SCRaMbLE system in directed evolution experiments. The composition and arrangement of chromosomes can be reshaped, resulting in different interaction preferences within loxPsym sites during each round of SCRaMbLE. By increasing the number of SCRaMbLE rounds, variants with enhanced fitness can be enriched.

Previous studies have demonstrated the potential of SCRaMbLE for directed evolution[21,23,24]. The SCRaMbLE experiment typically involves inducing Cre recombinase in yeast cells containing one or more synthetic chromosomes to generate a population with genomic heterogeneity. This population is then subjected to selective growth conditions for the screening of viable cells[52]. Compared with classical adaptive laboratory evolution via causal mutations, SCRaMbLE

facilitates rapid and extensive genetic rearrangements, including deletion, inversion, and duplication. In addition, aneuploidy is also considered as a quick response to stress[53]. Our findings suggested that SCRaMbLE based on the loxPsym sites across the 16 chromosomes in SparLox83R led to the potential occurrence of aneuploidy, such as whole chromosome deletion and duplication (Fig. 5c), which were rarely observed in cells with synthetic chromosomes. Thus, these two types of SCRaMbLE could complement each other and be orthogonal to other directed evolution methods as they provide different types of genetic changes[54]. However, further investigation is necessary to determine whether SCRaMbLE could provide a more rapid response to stress than classical adaptive laboratory evolution.

In summary, we used a strategy of random incorporation of loxPsym sites across all yeast chromosomes to develop SparLox83R, a strain with 83 loxPsym sites, and demonstrated the advantages of rapid environmental adaptation using the SCRaMbLE system. When construction of the final Sc2.0 yeast strains harboring synthetic chromosomes is complete and used with SparLox83R, an increased frequency of genomic rearrangements between SparLox83R-derived chromosomes and synthetic chromosomes can be expected in the hetero-diploid strain. In addition, more loxPsym sites can be incorporated into SparLox83R, further increasing its rearrangement capacity and improving its utility in a broad range of applications.

## Methods

### Strains
The yeast strains used in this paper were derivatives of BY4741/BY4742. Standard methods for yeast culture and transformation were applied. The strains generated in this study are listed in Supplementary Data 5.

### Consecutive insertion of loxPsym sites
SparLox83 was generated using CRISPR/Cas9-mediated multiplex DNA integration as previously described[55]. Briefly, three donor-gRNA cassettes were amplified and co-transformed into yeast cells alongside linearized plasmid (200 ng each) using the lithium acetate transformation method. Two plasmids were used, containing either *LEU2* or *URA3*. Insertions were verified by PCR. Colonies containing repaired donor-gRNA-containing plasmids were eliminated by streaking onto YPGal plates (2% bacto-peptone, 1% yeast extract, 2% galactose, and 2% agar). Colonies were subsequently streaked onto SC-Ura, SC-Leu, and YPD plates to confirm loss of plasmid. Verified strains were used for the next round of loxPsym insertion.

### Yeast colony PCR
Cells were resuspended in 20 μL of 20 mM NaOH and lysed in a thermocycler (94 °C, 3 min; 4 °C, 2 min, 5 cycles). Cell lysate (1 μL) was used as template for PCR using a site-specific forward primer upstream of each insertion site and a common reverse primer, MGO001 (Supplementary Data 4). PCR was performed using *EasyTaq* DNA polymerase (TransGen Biotech) and the conditions were as follows: 94 °C for 5 min; 30 cycles of 94 °C, 30 s; 55 °C, 30 s; 72 °C, 1 min; and final extension at 72 °C for 7 min. Agarose gel analysis was used for visualization of PCR products.

### Preparation of yeast genomic DNA
Yeast genomic DNA was prepared as previously reported[56] with minor modifications. Briefly, collected yeast cells were washed once in sterile water and resuspended in 100 μL breaking buffer (50 mM Tris pH 8.0, 100 mM NaCl, 1% SDS, 2% Triton X-100, and 1 mM EDTA) before addition of an equal volume of glass beads (Sigma G8772) and 200 μL of Phenol-Chloroform-Isoamyl alcohol (PCI) (25:24:1). After vortexing for 10 min at room temperature, 100 μL sterile water was added and mixed by inverting the tube several times. After centrifugation at $12,000 \times g$ for 10 min, the upper layer (150 μL) was transferred to a new

microfuge tube. For DNA precipitation, 500 μL 100% ethanol was added and the sample was incubated at −20 °C for 20 min. Genomic DNA was pelleted at 12,000 × $g$ for 5 min at 4 °C. The DNA pellet was dried at 45 °C for 15 min in vacuum centrifugal concentrator (Eppendorf Concentrator Plus) and then resuspended in 50 μL sterile water.

### Serial dilution assay

Yeast cells were cultured overnight in YPD medium with rotation at 30 °C. Cultures were adjusted to the same cell concentrations after measuring the optical density at 600 nm ($OD_{600}$) and then serially diluted 10-fold with water. Diluted cells were spotted onto selective media plates, incubated at 30 °C for 48–72 h, and then imaged for analysis.

### Assay for genomic stability

Yeast colonies were inoculated into 5 mL YPD medium and incubated at 30 °C for 24 h with agitation. Cultures (5 μL) were transferred into new vessels containing 5 mL fresh YPD medium and cultured under the same conditions. These steps were repeated daily for ten days (~120 generations), after which cells were streaked onto YPD plates and incubated at 30 °C for 1–2 days until colonies could be visualized. Single clones were isolated and subjected to PCR analysis to examine the presence of loxPsym sequences. In total, 35 independent clones were tested at 24 randomly chosen loxPsym sites.

### Pulsed-field gel electrophoresis

Chromosomal DNA was prepared as previously described[57]. In brief, about $5 \times 10^7$ cells at the stationary phase were collected and mixed with a 0.6% low melting point agarose solution before being pipetted into a casting mold. The solidified inserts were then subjected to digestion using zymolyase and proteinase K enzymes. After washing, the samples were analyzed on a BioRad CHEF Mapper apparatus using a 0.9% agarose gel for 20 h at 14 °C under an applied voltage of 6 V/cm and an angle of 120°, with switch times ranging from 10 to 60 s.

### SCRaMbLE of yeast cultures

SparLox83R or the heterozygous diploids with ReSCuES sequence integrated were transformed with pGAL-Cre plasmid, which was constructed by integrating the NatMX cassette into the LYS2 coding sequence based on a previously described plasmid[29]. Single yeast colonies were inoculated into 5 mL YPD medium containing 0.1 mg/mL clonNAT and cultured overnight at 30 °C. Cultures were diluted to $OD_{600} = 0.3$ in 5 mL fresh YEP medium with 2% w/v raffinose and 0.1% w/v glucose and cultivated at 30 °C for 6 h. Galactose was added to a final concentration of 2% w/v and cultures incubated at 30 °C for 2 h. Cultures were collected and plated onto solid YPD medium containing a selective agent.

For consecutive induction, five rounds of SCRaMbLE were induced as follows. SCRaMbLE was induced by the addition of galactose as described above. The single yeast colony was inoculated overnight and then transferred to fresh YEP medium with containing 2% w/v raffinose and 0.1% w/v glucose to cultivate for 6 h. Galactose was added to achieve a final concentration of 2% w/v, and the culture was incubated at 30 °C for an additional 2 h. Subsequently, cells were collected, washed once with 1 mL sterile water, resuspended in either liquid SC-Leu or SC-Ura medium in alternative rounds, and cultivated to the stationary phase. After five rounds of SCRaMbLE, the cells were plated on a SC-Leu plate, from which nine colonies were selected for sequencing.

### Split-URA3 mediated rearrangement

The URA3 gene was split in half by the intron of ACT1. A loxP sequence was inserted into the intron of ACT1. To integrate the two halves of the URA3 gene into the genome, a His marker was fused to one half of the URA3 gene and a Leu marker was fused to another half by PCR. Using transformation-associated recombination, the two halves of the URA3 gene with different markers were inserted into the location of XIV-1 and IV-5 of JDY524, respectively. A colony growing on the SC-His-Leu plate was picked and transformed with pGAL-Cre plasmid. The Cre expression was induced by the addition of galactose as described above. Put the inducted culture onto the SC-Ura plate and incubate at 30 °C for 48 h. Colonies were verified by PCR.

### Screening of HAc tolerance strains by continuous evolution

JDY544 and JDY546 were transformed with pGAL-Cre plasmid (clonNAT resistance) and several rounds of SCRaMbLE-assisted evolution were performed as follows. Each strain was evolved in 48 individual populations. Single colonies were cultured overnight in YPD containing 0.1 mg/mL clonNAT at 30 °C and re-inoculated to obtain an $OD_{600} = 0.3$ in YEP medium with 2% w/v raffinose and 0.1% w/v glucose. After growing for 6 h at 30 °C, galactose (2% w/v final concentration) was added and cultures incubated at 30 °C for an additional 2 h to induce Cre expression. To inhibit Cre expression, glucose (2% w/v final concentration) was added and cultures incubated at 30 °C until saturated. The SCRaMbLEd populations were diluted 1:10 and 1:100 into 100 μL fresh YPD. Diluted cells were plated on YPD agar plates and stress plates with series concentration of HAc. The strains incubated at 30 °C for several days until visible colonies were formed for all the 96 populations on stress plates with low concentration of HAc. For the next round of evolution, the parent strains were picked from the stress plate with visible colonies for all the 96 populations. Repeat the incubation and induction processes.

### Quantification for the growth of the evolved colonies

The growth of the strains after multiple rounds of directed evolution were quantitatively evaluated on YPD agar plates and stress plates with series concentration of HAc using CellProfiler v4.2.6 software. Given that the edges of the plates were relatively thick with no neighboring colonies, creating a nutrient-rich environment conducive to faster strain growth, colonies located at these plate edges were initially excluded. For the remaining colonies, a customized yeast patch identification workflow (https://cellprofiler.org/examples) were utilized to quantify the mean intensity of their forced spots. Specifically, we compressed the images into ~1200 × 800 png files and then cropped them based on colony positions in the images. The "Block size" parameter in the CorrectIlluminationCalculate step was set to 80, while for IdentifyPrimaryObjects step, we adjusted the "Typical diameter of objects" parameter to range between 8–80 units. In addition, in FilterObjects step, we modified both "Minimum FormFactor value" and "Minimum Area value" parameters to be 0.40 and 90, respectively. Finally, we conducted an unpaired two-tailed t-test to determine statistical significance regarding differences in growth.

### Whole-genome Nanopore sequencing

DNA extraction and sequencing were performed as described previously[26]. Briefly, genomic DNA was extracted according to using a modified Qiagen Genomic-tip 100/g protocol with the Qiagen Genomic Buffer kit. DNA quality was assessed with agarose gel electrophoresis and with a NanoDrop™ 2000 Spectrophotometer and a Qubit 4 Fluorometer with dsDNA HS reagent. Oxford Nanopore Technology (ONT) sequencing library preparation was performed using SQK LSK108 or LSK109 kits with Native Barcoding kits EXP-NDB104 or EXP-NBD114, according to the manufacturer's guidelines. Sequencing was performed on a MinION Mk1B device using FLO-MIN106D with R9.4.1 sequencing cells. Sequencing was performed for 48 h using MinKNOW (v3.6.5) software.

### Junction detection of SCRaMbLEants

All possible combinations of the 83 loxPsym sites were assembled in silico using the yeast reference genome (GCF_000146045.2). Only 1 kb-

long loxPsym neighboring sequences were retained. The assembled sequences were used as the reference for junction detection. LoxPsym-containing ONT sequence reads were identified using LAST software (v1250) and then aligned to the reference using Minimap2 (v2.20-r1061). Reads with MapQ > 20 that covered at least one reference sequence were retained for subsequent analysis. Reads containing more than one loxPsym site were analyzed separately. The rearrangement rate (RR) of a particular loxPsym site, i, was defined as $Nre_i/(2Nnorm_i + Nre_i)$, where $Nnorm_i$ and $Nre_i$ were the number of normal and rearranged reads for loxPsym site i. The Rearrangement Rate (RR) was subsequently normalized to the total number of rearranged loxPsym sites to calculate Average Rearrangement Rate (ARR). The rearrangement weight (RW) of two particular loxPsym sites, i and j, was calculated by Eq. 1, which also considered bias of sequencing depth.

$$RW_{i,j} = \frac{Nre_{ij}}{\sqrt{Nall_i * Nall_j}} \tag{1}$$

$Nre_{ij}$ is the count of rearranged reads between loxPsym sites i and j, and $Nall_i = 2Nnorm_i + Nre_i$. Networks of inter-chromosomal and intra-chromosomal rearrangements were constructed using Cytoscape software (v3.7.1).

## Assembly-based structural variant calling

ONT reads (FASTQ format) were used with the Canu pipeling (v2.1.1) to assemble the SparLox83R and JDY528 genomes. Subsequently, Pilon software was employed to correct draft assemblies, utilizing the high-quality Illumina PE300 reads, for three consecutive rounds. The assembled genomes of JDY528 was aligned to SparLox83R using Minimap2 (v2.20-r1061) software with -x asm5 argument. The resultant PAF format files were used to display the structural variations.

## Structural and copy number variant and loss of heterozygosity detection in hetero-diploid strains

ONT reads (FASTQ format) from hetero-diploid strains were filtered and trimmed using NanoFilt software. LoxPsym-containing ONT reads were obtained by LAST software (v1250) for structural variant (SV) detection. To avoid alignment bias due to the similarities of ChrIII and SynIII, LAST was used to separate loxPsym-containing reads according to whether they contained SynIII PCRTags. Reads containing SynIII PCRTags were mapped to SynIII and were used as the inputs to Euclidean path algorithms, which required only linear time to find a sequence consistent with the loxPsym junctions observed in the reads, to reconstruct SCRaMbLEd SynIII as previously described[18]. Reads without SynIII PCRTags were aligned to the SparLox83R genome for SV calling using the NGMLR-sniffles (v1.0.12 for sniffles and v0.2.7 for ngmlr) pipeline. In addition, SynIII PCRTag-containing reads were aligned to the SparLox83R genome to detect translocation between SynIII and SparLox83R. LoxPsym sites that were not identified in any reads were considered to be the result of Loss of Heterozygosity (LOH). We used a previously described iterative algorithm to estimate and refine the copy number variation (CNV) of loxPsym sites[28]. Briefly, the sequencing depth of the flanking 2000bp of loxPsym sites was calculated for hetero-diploid strains and the depth of SCRaMbLEd strains were corrected according to the parental strain using the iterative algorithm.

## Hi-C library preparation and data processing

Hi-C library preparation was performed as previously described[58] at Beijing Novogene Bioinformatics Technology Co. Ltd. Samples, each with two biological repeatations, were flash-frozen and pulverized before formaldehyde cross-linking. Briefly, cell lysis and *Mbo*I digestion were performed after cross-linking, then DNA ends were labeled with biotin and joined. DNA was decrosslinked using Protease K and

SDS. Free DNA was purified and extracted using Ampure XP beads. After quality control, ultrasonication was used to produce 200–500 bp DNA fragments. The biotin-labeled fragments were further enriched and sequenced by Illumina Novaseq. Hi-C data were mapped to the public S288C reference genome (GCF_000146045.2) using distiller-nf (v0.3.3). Read pairs that were not uniquely mapped (mapping score <1) were discarded. Valid alignment files were transformed to .hic files for juicer using pre command. Hi-C map resolution was calculated as described previously[59]. Contact matrices used for heatmap visualization and further analysis were KR (Knight-Ruiz) normalized at 5 kb resolution. To compare the interaction frequencies between SparLox83R and other strains, the $\log_2$ ratio of a 5 kb binned contact map was computed and normalized by subtracting the median of its values.

For the construction of a 3D chromosome model, Hi-C data were mapped to the SparLox83R genome and binned at 5 kb resolution. Chromosomal 3D structures were then inferred using the Pastis (v0.4) method with a Poisson model and visualized using PyMOL.

## RNA sequencing and analysis

SparLox83R and JDY528 strains, each with 3 biological replicates, were cultured overnight in YPD medium at 30 °C. Subsequently, they were re-inoculated in 50 mL fresh YPD medium for further incubation until $OD_{600}$ reached -0.6. The cells were treated with DMSO or nocodazole at a concentration of 10 µg/mL for 2 h prior to harvesting. The RNA sequencing experiments were conducted at Beijing Novogene Bioinformatics Technology Co., Ltd. Libraries were prepared using the non-stranded Illumina NEBNext UltraTM RNA Library Prep Kit. Raw data adapters were removed using Cutadapter (v3.4) software. Cleaned data was mapped using Hisat2 (v2.2.1), and PCR duplicates were identified and eliminated using picard (v2.25.7). Gene read counts were generated by htseq-count software (v0.13.5) with the intersection-nonempty option, while $\log_2$(fold change) and padj values were calculated using DESEQ2 package (v1.30.1). TPM values for each gene were determined by Stringtie software (v2.1.7).

## Statistics and reproducibility

Unless specially noted, two-tailed t tests were employed to compare different groups, while simple linear repression was used to assess the correlation between two groups in this paper. Statistical significance was considered at a $p$-value < 0.05 threshold. *$p$ < 0.05, **$p$ < 0.01, ***$p$ < 0.001, ****$p$ < 0.0001. Hi-C experiments were independently replicated twice, and RNA-seq analysis included three biological replicates.

## Reporting summary

Further information on research design is available in the Nature Portfolio Reporting Summary linked to this article.

## Data availability

The sequencing data that support the findings of this study have been deposited into NCBI BioSample database with the BioProject accession number: PRJNA768098. The reference sequence of synIII used in this study can be downloaded from GenBank with accession codes KC880027. The complete dataset supporting the findings of this study can be found in both the main manuscript file and its Supplementary Information files. Source data are provided with this paper.

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

## Acknowledgements

This work was supported by the National Key R&D Program of China (2022YFF1201800, 2018YFA0900100), National Natural Science Foundation (32030004, 32150025, 32122050, 32101184, 32001042), Youth Innovation Promotion Association CAS (2021359), Guangdong Natural Science Funds for Distinguished Young Scholar (2021B1515020060), Guandong Basic and Applied Basic Research Foundation (2023A1515030285), Guangdong Provincial Key Laboratory of Synthetic Genomics (2023B1212060054), Bureau of International Cooperation, Chinese Academy of Sciences (172644KYSB20180022), Shenzhen Science and Technology Program (KQTD20180413181837372, KQTD20200925153547003), Innovation Program of Chinese Academy of Agricultural Science and Shenzhen Outstanding Talents Training Fund. This work was also supported by UK Biotechnology and Biological Sciences Research Council (BBSRC) grants BB/M005690/1, BB/P02114X/1, and BB/W014483/1, Royal Society Newton Advanced Fellowship (NAF\R2\180590) and a Volkswagen Foundation "Life? Initiative" Grant (Ref. 94,771) to Y.C. W.Z. and J.D.B. were supported by US NSF grants MCB-1026068, MCB-1443299, MCB-1616111, and MCB-1921641. We thank Gabrielle David for proofreading this manuscript.

## Author contributions

J.D. conceived and designed the study. J.D., C.H., Y.M., and Y.C. supervised the experiments. L.C., S.Z., T.L., S.H., Z.L., W.Y., S.J., M.M., D.S., and W.Z. performed the experiments. J.X., S.Z., and C.H. analyzed HiC data. J.D., L.C., S.Z., T.L., Z.L., Y.M., Y.C., and J.D.B. analyzed the data and wrote the manuscript. All authors have read and approved the final manuscript.

## Competing interests

J.D.B. is a Founder and Director of CDI Labs, Inc., a Founder of and consultant to Neochromosome, Inc, a Founder, SAB member of and consultant to ReOpen Diagnostics, LLC and serves or served on the Scientific Advisory Board of the following: Logomix, Inc., Modern Meadow, Inc., Rome Therapeutics, Inc., Sample6, Inc., Sangamo, Inc., Tessera Therapeutics, Inc., and the Wyss Institute. The remaining authors declare no competing interests.
