## [Peer Review File · Nature Communications]

Reviewers' Comments:

Reviewer #1:

Remarks to the Author:

The authors apply SCRaMBLE genome-wide in *Saccharomyces cerevisiae*. To make this experiment possible, they engineer a yeast strain containing 83 loxP sites distributed across all 16 nuclear chromosomes. They induce SCRaMBLE in this strain, as well as in an F1 produced by mating this strain to one containing synIII, a synthetic version of Chromosome III that also contains many loxP sites. These experiments show how SCRaMBLE can be applied on a genome-wide scale to generate genomic changes within and between chromosomes, producing strains with diverse genome structures. This system can help improve understanding of genome structure and its relationship to traits of evolutionary and biotechnological interest.

Comments:

1. In the Introduction's sentence beginning 'To overcome this problem', the wording could be a little more precise. Both I-SceI and CRISPR/Cas9 systems involve induced DSB repair. Only the first is stated as involving induced DSB repair.
2. In the following sentence, the word 'pre-determined' is used but arguably SCRaMBLE is still somewhat predetermined. In SCRaMBLE, recombination only happens where loxP sites are present.
3. In the Results section entitled 'Genome-wide insertion...', the following modification would help: 'Previous studies employing SCRaMBLE in strains with [multiple] synthetic chromosomes...'
4. In the Results section entitled 'Induction of Cre recombinase activity...', succinctly explain the ReSCuES system (a couple sentences or so). Otherwise, the reader must look at the previous paper to understand why ReSCuES is mentioned.
5. Figure 4a. I found the arrow notation insufficiently clear. Segments with very similar colors are indistinguishable. This impacts interpretation of SCRaMBLE strains, as it is hard to tell when a single segment has been duplicated multiple times as opposed to multiple neighboring segments having been duplicated. Perhaps in addition to the color gradient, using a mix of filled and unfilled arrows would help. There might be further options to increase the space of distinct arrows, such as crosshatching the fill.
6. In the Methods section entitled 'Split-URA3 mediated rearrangement', 'splitted' should be 'split'.
7. In the Methods section entitled 'Junction deletion...', '((' should be '('.
8. I might have missed them, but data and code availability statements could be missing.

Reviewer #2:

Remarks to the Author:

Li and coworkers report a new demonstration of the SCRaMBLE technology, where LoxPSym sites are integrated into the *S. cerevisiae* genome to yield controllable Cre-mediated chromosomal recombination events. Previous demonstrations of SCRaMBLE only involved LoxP sym sites integrated into one or a few (synthetic) chromosomes, which resulted in few inter-chromosomal rearrangements. In this new iteration of the technology, 83 LoxPSym sites were integrated in various positions distributed over all 16 *S. cerevisiae* chromosomes in the laboratory strain BY4741/s288c in an attempt to fuel more inter-chromosomal events.

The results show that the technology works and results in a broad range of intra- and inter-chromosomal recombination events between the different LoxPSym sites, with a marked increase in recombination frequency for sites that are located close to each other on the chromosome and sites located in open chromatin regions. The authors go on to demonstrate that recombination

followed by selection for increased stress tolerance can in some (but not all) cases (i.e. different stress factors) quickly result in the selection of novel variants that show increased tolerance.

The authors have conducted a significant amount of high-quality experimental and computational work. Moreover, the text is well-written and the figures are generally clear.

Despite perhaps not representing a major novel or innovative breakthrough compared to the previous papers that introduce and exploit SCRaMbLE, I feel that this study is interesting because it demonstrates directly just how important structural variation can be for quick evolutionary adaptation (an aspect that is perhaps not discussed in sufficient depth in the paper, see below) and opens new avenues to study this process in more detail in any strain, not just the Sc2.0 strains. Secondly, it provides an alternative route to employ SCRaMbLE to create novel, improved yeast variants.

Whereas I am generally enthused about the work, I also believe that adding a more detailed analysis of the current results and a validation of some obtained results would improve the impact of the manuscript.

Major comments

Some logical controls seem to be missing from the stress-adaptation experiments. As far as I can see, no wild-type cells (same strain background but not containing any LoxP sites) or uninduced SparLox83 (strain with the LoxP sites, but without induction of the Cre enzyme to induce recombination) were analyzed? This makes it difficult to estimate if and how the SCRaMbLE technology resulted in a quicker and/or better adaptation to stress, which in my mind is an essential piece of information to assess the usefulness of the technology for applications. Moreover, it would also be interesting to measure the potential fitness cost of the specific rearrangements that yield resistance to stress in non-stressful conditions (in other words: what is the tradeoff cost of the rearrangement) and compare this to resistant mutants obtained without SCRaMbLE. I also realize that this last suggestion might be outside the direct scope of this study, although I do think it would be a very interesting aspect that would allow us to better understand the potential of SCRaMbLE.

Why was the adaptive laboratory evolution experiment not carried out in the same way (using the same stress factors) for the haploid and diploid strains? It would be interesting to be able to compare the adaptation between these two starting strains...

I do not agree with the following sentence: "These data indicate that LoxP-mediated large-scale genomic rearrangements changed the 3D genome structure, generating an advantageous phenotype". I do not think that the data provide solid proof for the causative nature of the observed 3D-rearrangements. It is for example equally possible that changes in local chromatin structure (independent of changes in nuclear organization), readthrough of new neighboring genes, or copy number variation result in beneficial changes in expression of key genes.... In fact, I was surprised to see a Hi-C analysis, but no transcriptome analysis. I do not mean to suggest that it is essential for the authors to tease out what is causing the increased resistance, but I do think it is best to lower the claims and provide a better discussion of all factors that could play a role in the link between the rearrangements and the phenotypes.

Minor:

1. Introduction: it would be interesting to give a better overview of the different SCRaMbLE techniques, and also already mention why few inter-chromosomal rearrangements have been found and analyzed in these previous SCRaMbLE studies (namely because sites were often only present on one chromosome, and/or because recombination favored sites that are located close to each other, and those were always present in the dense arrays of LoxP sites in the Sc2.0 project...)
2. Should this technique still be called SCRaMbLE "Synthetic Chromosome Rearrangement and Modification by LoxPsym-mediated Evolution", since it does not involve synthetic chromosomes anymore?

3. Results: It could be a good idea to re-evaluate the labeling of the strains. More informative names (instead of long, unintelligible codes) would avoid confusion and make the manuscript more understandable. Perhaps most importantly, consider re-naming JDY525, as it is the strain that contains 83 Lox sites, and SparLox83 being the code name for the same strain that now also harbors the ReSCuES system. A more systematic, uniform naming would make it easier for the reader to follow. In addition, please indicate that JDY525 was constructed from strain BY4741 (this strain only pops up later in the text).

4. Results: Briefly explain the ReSCuES system.

5. In the results sections, the authors mention statistical terms in the main text for observations for which no statistical tests were performed, for example: 'no correlation' (Extended Data Fig.5b), 'significant' (Extended Data Fig.3b, Extended Data Figs. 8b&c).

6. Summary: The claim that SparLox83xSynIII led to increased genomic rearrangements is not justified by the results; increased compared to which strain? SparLox83 and SynIII were not tested by themselves here.

7. Summary: "analysis of these evolved strains..."; needs rephrasing, the authors refer to the wrong experiment (SparLox83xSynIII instead of SparLox83).

8. Section 1: Construction of 83 genomic edits within one strain is very impressive and interesting. It would be very helpful to the field to gain insight in the number of unintended SNPs/InDels that were accumulated during strain construction.

9. Section 1: It is confusing to me that strain JDY525 is used for these experiments, as the rest of the study is based on SparLox83. I suggest to add the results for this strain to Fig.1b.

10. Section 1: Please add more information on the design of SparLox83: how were the insertion sites of LoxP sites chosen? Why were they not equally distributed amongst the chromosomes? Were they consistently inserted in 3'UTRs (as in the Sc 2.0 project) or also within ORFs/5'UTRs/...?

11. Fig.1a: Only 74 LoxP sites are depicted, please add all sites.

12. Fig. 1c: Could the authors provide the evidence for these claims as supplemental figure?

13. Section 2: The authors stress that inter-chromosomal rearrangements pre-dominate over intra-chromosomal rearrangements. Although this is true for SparLox83, I wonder whether it is a fair general statement to make? In fact, intra-chromosomal rearrangements are more likely to occur (an observation which is also made in Fig. 2d and in section 6 (Table S6)), no?

14. Section 3: It is unclear how 'chromatin accessibility' was defined/determined, this information should be added somewhere.

15. Although I am convinced that the differences in recombination frequency that are observed between different LoxP sites are indeed mostly due to 1) their genomic distance and 2) the local chromatin structure, one cannot exclude other confounding factors, such as for example bias due to selection. Even in rich medium like YPD and optimal growth conditions, some rearrangements might cause a severe fitness defect or even cell death.... This could be pointed out more explicitly.

16. Fig. 2e: I suggest adding the information for intra-chromosomal ARR.

17. Section 4: The data for growth on ethanol, H₂O₂ and NaCl are missing from Extended Data Fig. 6a, please provide this information.

18. Section 5: Could the authors analyze and describe the results in more detail? Information on how many recombination events were observed in total, how many per strain, how many inter-/intra-chromosomal and how many on synIII, how many of each type of recombination, is not mentioned, but valuable info. I believe some of this data is given in Table S6, but I would strongly

encourage the authors to add an extra panel to Fig. 4 to discuss this information and adapt their interpretation of the results.

19. Fig. 4a: Define y-axis, as the figure shows the structural rearrangements in synIII, rearrangements between synIII and chromosomes from SparLox83 should also be shown in this panel.

20. Fig. 5a: not very clear, please specify why some reads are grey/blue and add ChrA/B to all schematics.

21. Fig. 5b: why is this data not shown for all chromosomes? Could the authors provide this information as supplemental material? And, add the data for ChrIII to the main figure?

22. Section 7: not all populations show growth media with 0.4 % HAc (Extended Data Fig. 8b), please adapt the text.

23. Discussion: It is very interesting that more recombination events were observed in haploid strains compared to the diploid. However, the first explanation of the authors (that recombination in haploids might lead to lethality) actually suggests the opposite. The second explanation should be rephrased (it is not the same experiment) and could in principle be investigated (the data is available to look for LOH and aneuploidy in these strains). Could you hypothesize more on potential explanations and mention whether this has been observed previously?

24. Discussion: I am confused about the mentioning of a haploid strain with an essential gene plasmid. I believe this was not mentioned in the result section, could you please clarify this?

25. Discussion: At the end of the section, the authors mention that an increased frequency of genomic rearrangements between SparLox83-derived chromosomes and synthetic chromosomes can be expected. I wonder why this is claimed, as the data in the manuscript suggests otherwise (e.g. the general decrease in genomic rearrangements for diploid strains (fig.1) and the low frequency of SparLox83 rearrangements in the presence of other synthetic chromosomes (Table S6))?

26. Discussion: you mention that chromosomal rearrangements (or more commonly: structural variation) is a known, important driver of adaptive evolution. This is especially the case in yeast, where genomic rearrangements are often hypothesized to offer a "quick fix" to a sudden, harsh challenge (as they are often used in ALE). It would perhaps be appropriate to cite some of the pioneering work here, and discuss in more depth how SCRaMbLE could be used to study this process in more detail.

27. Discussion: Please discuss if and how you could overcome the huge preference for recombination between proximal Lox sites in the complete Sc2/.0? If this is not resolved, the repertoire of rearrangements that will result from recombining the SCRaMbLE system in Sc2.0 will be quite limited?

28. Methods, yeast colony PCR: indicate polymerase

29. Methods, SCRaMbLE: provide information on the Cre-plasmid (Addgene ID/reference), specify if cells were kept in liquid cultures for consecutive rounds of SCRaMbLE or if they were plated in between.

30. Data availability statement is missing and access of the WGS data is not provided.

31. Table S2: specify the locus of ReSCuES integration in SparLox83.

Typos:

1. Summary: heterodiploid strains < strain; strains possessing synIII < strain
2. Results, section 2: $r=0.9828$ >< fig.2c mentions R2

3. Results, section 4: reference Luo et al., 2018 < reference 26
4. Results, section 5: 'often' deleted: often should be removed if all 9 colonies were obtained from a single induced population. It should be mentioned clearly in the text how many biological repeats were used. If this was one biological repeat, also the discussion 'similar biases' should be adapted.
5. Results, Fig.5: bule < blue
6. Results, section 5: mention LOH=loss of heterozygosity
7. Discussion, were existed < existed
8. Methods, SCRaMbLE: SparL83<SparLox83
9. Methods, Split-URA3: palte<plate, inducted<induced, other typo's
10. Methods, WGS: according<according
11. Methods, junction detection: number 2 seems to be a mistake in formula $N_{rei}/2N_{ormi}+N_{rei}$
12. Methods, Hi-C library: Spar83L<SparLox83

REVIEWER COMMENTS

Reviewer #1 (Remarks to the Author):

The authors apply SCRaMBLE genome-wide in *Saccharomyces cerevisiae*. To make this experiment possible, they engineer a yeast strain containing 83 loxP sites distributed across all 16 nuclear chromosomes. They induce SCRaMBLE in this strain, as well as in an F1 produced by mating this strain to one containing synIII, a synthetic version of Chromosome III that also contains many loxP sites. These experiments show how SCRaMBLE can be applied on a genome-wide scale to generate genomic changes within and between chromosomes, producing strains with diverse genome structures. This system can help improve understanding of genome structure and its relationship to traits of evolutionary and biotechnological interest.

Response: We appreciated the reviewer for recognizing the values and significance of our work and providing the helpful comments. The point-to-point responses are as follows.

Comments:

1. In the Introduction's sentence beginning 'To overcome this problem', the wording could be a little more precise. Both I-SceI and CRISPR/Cas9 systems involve induced DSB repair. Only the first is stated as involving induced DSB repair.

Response: Thank you for pointing this out. We have revised the text as follows: "To quantify the impact of genomic rearrangement independently, several methods have been developed to construct targeted long-range rearrangements in model organisms, including, Cre/loxP recombination, *I-SceI*-induced and CRISPR/Cas9-induced DSB repair."

2. In the following sentence, the word 'pre-determined' is used but arguably SCRaMBLE is still somewhat predetermined. In SCRaMBLE, recombination only happens where loxP sites are present.

Response: We thank for the reviewer's careful reading. We agree with the reviewer's comment and have deleted the word 'pre-determined'.

3. In the Results section entitled 'Genome-wide insertion...', the following modification would help: 'Previous studies employing SCRaMBLE in strains with [multiple] synthetic chromosomes...'

Response: We appreciate the reviewer for the comments and have revised the sentence according to the reviewer's suggestion.

4. In the Results section entitled 'Induction of Cre recombinase activity...', succinctly explain the ReSCuES system (a couple sentences or so). Otherwise, the reader must look at the previous paper to understand why ReSCuES is mentioned.

Response: We appreciate the reviewer for pointing this out. We have explained the ReSCuES system in the revised manuscript as follows: "ReSCuES was a reporter to efficiently identify SCRaMBLEd cells based on a loxP-mediated switch of "on" and "off" states of the two auxotrophic markers, *URA3* and *LEU2*."

5. Figure 4a. I found the arrow notation insufficiently clear. Segments with very similar colors are

indistinguishable. This impacts interpretation of SCRaMBLE strains, as it is hard to tell when a single segment has been duplicated multiple times as opposed to multiple neighboring segments having been duplicated. Perhaps in addition to the color gradient, using a mix of filled and unfilled arrows would help. There might be further options to increase the space of distinct arrows, such as crosshatching the fill.

Response: We are sorry for the confusing writing in the manuscript. The arrow notation with gradient rainbow colors was followed the “SCRaMbLEgrams” which was a tool to depict the recombination events observed in SCRaMbLEd strains in Sc.2.0 project (*Genome Research*, 2016, PMID: 26566658). In response to the reviewer’s suggestion, we have regenerated the graph (Figure RL1a). However, accurately discerning the precise location of each segment within the rearranged chromosome remains challenging. Nevertheless, it was evident from both the previous and modified graphs that each strain exhibited a unique synIII structure. Considering no significant improvements were achieved and to maintain consistency with other studies in the Sc.2.0 project, this graph has not been revised accordingly.

In addition, detailed information can be found in Extended Data Fig. 11a&b. Taking JDY553 and JDY596 as examples, the segments of the rearranged genome were plotted against the corresponding segments of the parental genome in their original sequential order (Figure RL1b). As visualization, rearrangement types were classified as deletions, inversions, duplications, duplicated inversions, inverted duplications, and complex rearrangements with respect to the parental synIII sequence. Then, the fate of each segment in each strain could be determined (Figure RL1c). By combining this information together, a comprehensive depiction of synIII's specific rearrangement could be achieved.

Figure RL1. **a**. The arrangement of synIII in each strain modified according to the reviewer's suggestion. **b**. Dot-plots illustrating synIII rearrangements in JDY553 and JDY596. Each dot represents a fragment between two loxP sites. Black dots represent fragments arranged in synIII sequence order and blue dots represent fragments inverted with respect to the synIII sequence. **c**. The fate of each segment in each strain was classified as deletion (gray), inversion (blue), duplication (yellow), duplicated inversion (dark green), inverted duplication (light green), multiple duplication (red) or identical to the parent strain (white).

6. In the Methods section entitled 'Split-URA3 mediated rearrangement', 'splitted' should be 'split'.
 Response: As suggested, we have changed the "splitted" to "split".

7. In the Methods section entitled 'Junction deletion...', '((' should be '(.'
 Response: We sincerely thank the reviewer for careful reading and this has been corrected.

8. I might have missed them, but data and code availability statements could be missing.

Response: Thanks for pointing this out and we have supplemented data availability statement in the revised manuscript.

Reviewer #2 (Remarks to the Author):

Li and coworkers report a new demonstration of the SCRaMbLE technology, where LoxP sites are integrated into the *S. cerevisiae* genome to yield controllable Cre-mediated chromosomal recombination events. Previous demonstrations of SCRaMbLE only involved LoxP sites integrated into one or a few (synthetic) chromosomes, which resulted in few inter-chromosomal rearrangements. In this new iteration of the technology, 83 LoxP sites were integrated in various positions distributed over all 16 *S. cerevisiae* chromosomes in the laboratory strain BY4741/s288c in an attempt to fuel more inter-chromosomal events.

The results show that the technology works and results in a broad range of intra- and inter-chromosomal recombination events between the different LoxP sites, with a marked increase in recombination frequency for sites that are located close to each other on the chromosome and sites located in open chromatin regions. The authors go on to demonstrate that recombination followed by selection for increased stress tolerance can in some (but not all) cases (i.e. different stress factors) quickly result in the selection of novel variants that show increased tolerance.

The authors have conducted a significant amount of high-quality experimental and computational work. Moreover, the text is well-written and the figures are generally clear.

Despite perhaps not representing a major novel or innovative breakthrough compared to the previous papers that introduce and exploit SCRaMbLE, I feel that this study is interesting because it demonstrates directly just how important structural variation can be for quick evolutionary adaptation (an aspect that is perhaps not discussed in sufficient depth in the paper, see below) and opens new avenues to study this process in more detail in any strain, not just the Sc2.0 strains. Secondly, it provides an alternative route to employ SCRaMbLE to create novel, improved yeast variants.

Whereas I am generally enthused about the work, I also believe that adding a more detailed analysis of the current results and a validation of some obtained results would improve the impact of the manuscript.

Response: We appreciated the reviewer for recognizing the values and significance of our work and providing the constructive and insightful comments. The point-to-point responses are as follows.

Major comments

Some logical controls seem to be missing from the stress-adaptation experiments. As far as I can see, no wild-type cells (same strain background but not containing any LoxP sites) or uninduced SparLox83 (strain with the LoxP sites, but without induction of the Cre enzyme to induce recombination) were analyzed? This makes it difficult to estimate if and how the SCRaMbLE technology resulted in a quicker and/or better adaptation to stress, which in my mind is an essential piece of information to assess the usefulness of the technology for applications. Moreover, it would also be interesting to measure the potential fitness cost of the specific rearrangements that yield resistance to stress in non-stressful conditions (in other words: what is the tradeoff cost of the rearrangement) and compare this to resistant mutants obtained without SCRaMbLE. I also realize that this last suggestion might be outside the direct scope of this study, although I do think it would

be a very interesting aspect that would allow us to better understand the potential of SCRaMbLE. Response: Thank you for the reviewer's comments. Regarding the absence of control cells, there are two possible explanations. Firstly, it should be noted that the SCRaMbLE system is an inherent feature of Sc 2.0 and its effectiveness in facilitating adaptive evolution has been demonstrated in previous studies. Comparative colonies with increased ethanol tolerance were observed when β -estradiol was added to induce Cre recombinase expression, while no colony growth was observed on stress plates without β -estradiol induction (Figure RL2a from *Nature Communications*, 2018, PMID:29789540). Similar results have also been reported in other synthetic chromosome-based SCRaMbLEing studies. Therefore, we believe that the induction of Cre is necessary to generate strains with enhanced HAc tolerance.

Secondly, our aim in conducting stress-adaptation experiments was to demonstrate that genomes containing sparsely distributed loxPsym sites could serve as a potential strategy for enhancing strain performance under stressful conditions. To achieve this goal, we employed two different genotypic strains: SparLox83R \times synIII and BY4741 \times synIII. Both strains contained a synthetic chromosome and one copy of 16 chromosomes with or without loxPsym sites, respectively. The diploid strain BY4741 \times synIII was used as a control because previous studies have shown that heterozygous diploid strains carrying a synthetic chromosome can rapidly generate new phenotypes through environmental selection (Figure RL2b from *Nature Communications*, 2018, PMID:29789590). After three rounds of evolution, populations derived from SparLox83R \times synIII exhibited superior growth on media containing 0.6% HAc compared to those derived from BY4741 \times synIII. Considering the genotypic differences between these two strains, we concluded that the accelerated rate of evolution may be associated with rearrangements mediated by sparsely distributed loxPsym sites.

In addition, thank you for the later suggestion, we would be pursued late.

Figure RL2. **a.** SynXII strain with or without inducing SCRaMbLE by estradiol was plated onto SC-Ura plates containing 8% of ethanol. The cited figure was derived from the article PMID:29789540. **b.** Serial dilution assay comparing the growth of SCRaMbLEd strains to their non-SCRaMbLEd parent (yMS521) on high caffeine YPD plates. The cited figure was derived from the article PMID:29789590.

Why was the adaptive laboratory evolution experiment not carried out in the same way (using the same stress factors) for the haploid and diploid strains? It would be interesting to be able to compare the adaptation between these two starting strains...

Response: Thank you for the reviewer’s comments. We have indeed conducted a similar experiment using the haploid and diploid strains. Initially, we tested the original strains, SparLox83R and JDY536, for HAc tolerance (Figure RL3a). Subsequently, 12 individual colonies from each strain were inoculated into a 96-well microplate, and SCRaMbLE-Selection based evolution cycles were performed. Following the first round of SCRaMbLE (Round 1), three different cell concentrations ranging from OD₆₀₀ 1.0 to 0.01 were tested on YPD plates with a gradient of HAc (Figure RL3b). Compared to the parental strains, both haploid and diploid populations exhibited increased tolerance to HAc. However, the diploid populations demonstrated superior growth compared to those derived from SparLox83R. After the second round, most SCRaMbLEant populations of the diploid strains were able to grow on 0.6% HAc media at an OD₆₀₀ =1.0, while populations from SparLox83R still displayed growth defect on 0.5% HAc media at an OD₆₀₀ =0.1 and 0.01 (Figure RL3c). Further evolution was not pursued as the diploid populations exhibited better growth than those derived from SparLox83R.

Although a differential evolution rate was observed between the haploid and diploid strains, we aimed to investigate the potential of enhancing strain performance under stressful conditions by combining genomes containing sparsely distributed loxPsym sites and synthetic chromosomes, which theoretically could generate greater genomic heterogeneity. Therefore, these results were not included in the revised manuscript.

Figure RL3. Rapid development of HAc tolerance using SCRaMbLE in the haploid and diploid strains. **a.** Serial dilution assay on YPD medium with/without HAc. Cells at log phase were diluted to OD₆₀₀ =1.0, and then 10-fold dilution. Two strains, SparLox83R and JDY536, showed growth defect on YPD with 0.4% and 0.5% HAc. **b&c.** Growth of 24 SCRaMbLEant populations after first and second round SCRaMbLE. After SCRaMbLE, the saturated culture was diluted to OD₆₀₀ =1.0, 0.1 or 0.01 and spotted onto YPD with/without HAc.

I do not agree with the following sentence: “These data indicate that LoxP-mediated large-scale genomic rearrangements changed the 3D genome structure, generating an advantageous phenotype”. I do not think that the data provide solid proof for the causative nature of the observed 3D-rearrangements. It is for example equally possible that changes in local chromatin structure

(independent of changes in nuclear organization), readthrough of new neighboring genes, or copy number variation result in beneficial changes in expression of key genes.... In fact, I was surprised to see a Hi-C analysis, but no transcriptome analysis. I do not mean to suggest that it is essential for the authors to tease out what is causing the increased resistance, but I do think it is best to lower the claims and provide a better discussion of all factors that could play a role in the link between the rearrangements and the phenotypes.

Response: Thank you for the reviewer to point this out. According to the reviewer's suggestion, we have revised the sentence to "These data revealed a connection among large-scale rearrangement, genome 3D structure, transcription, and phenotype, demonstrating that simply reconfiguring chromosome architecture was sufficient to provide fitness advantages in stressful growth conditions."

In addition, we performed the transcriptome analysis comparing JDY528 and SparLox83R. As shown in Figure RL4a, the duplicated region (from *YDR073W* to *YDR277C*) exhibited significant up-regulation of the majority of genes. Conversely, a decrease in copy number from two to one on chrIX resulted in decreased expression levels for most genes. The differentially expressed genes were further mapped to the 3D model of JDY526 genome and showed strong colocalization with 3D structure alterations. Only one gene, *YDR001C*, located nearby the junctions, showed differential expression. However, its knock-out strain showed comparable tolerance to nocodazole when compared to wild-type strain (Figure RL4b). Considering that changes in expression levels were not limited to neighboring regions of the translocation breakpoint, it may explore a larger set of genes in subsequent analyses. Furthermore, it was possible that nocodazole resistance could be influenced by both the large-scale duplication on ChrIV and the copy number variation of ChrIX. The information has been added in the revised Extended Data Fig. 10.

Figure RL4. Transcriptome analysis between JDY528 and SparLox83R. **a.** Volcano plot of global expression changes of JDY526 compared to SparLox83R. **b.** Serial dilution assay comparing the knock-out strain and the wild-type strain.

Minor:

1. Introduction: it would be interesting to give a better overview of the different SCRaMbLE techniques, and also already mention why few inter-chromosomal rearrangements have been found and analyzed in these previous SCRaMbLE studies (namely because sites were often only present on one chromosome, and/or because recombination favored sites that are located close to each other, and those were always present in the dense arrays of LoxP sites in the Sc2.0 project...)

Response: We appreciate the reviewer for the constructive comments. Following the reviewer's suggestions, we have supplemented an overview of the different SCRaMbLE techniques in the introduction section as follows:

“Moreover, intra-chromosomal rearrangements dominate, with inter-chromosomal rearrangements rarely detected in individual SCRaMbLEd strains despite the utilization of a strain with four synthetic chromosomes³¹. By sequencing SCRaMbLEd pools derived from inducing a strain harboring 5.5 synthetic chromosomes, comparable frequencies of intra- and inter-chromosomal events were detected at 47.24% and 52.67%, respectively³². However, only 9.87% of the total reads accounted for inter-chromosomal recombination, indicating that the occurrence of inter-chromosomal recombination was significantly lower than that of intra-chromosomal recombination in cells containing multiple synthetic chromosomes. Considering that the spatial proximity between loxPsym sites plays a crucial role in chromosome interactions, it can be inferred that translocation events would be more frequent when loxPsym sites are distributed throughout the entire genome.”

Please see line 86-96 in page 5.

2. Should this technique still be called SCRaMbLE “Synthetic Chromosome Rearrangement and Modification by LoxPsym-mediated Evolution”, since it does not involve synthetic chromosomes anymore?

Response: Thank you for the reviewer to point this out. We agree with the reviewer that SCRaMbLE is a synthetic chromosome engineering technique which was first introduced as part of the Sc2.0-project. It is based on Cre/loxPsym recombination system by integration of loxPsym sequences into synthetic chromosomes. As the Cre/loxPsym recombination system is a universal and functional way to generate structural variation, the general concept of SCRaMbLE can be exploited to other yeast strains equipped with recombination sites (*Nature Communications*, 2018, PMID:29789533). In this study, we employed a CRISPR/Cas9-based strategy to efficiently integrate loxPsym sequences into multiple genomic loci, thereby inducing structural variations across the entire genome through the Cre/loxPsym recombination system. Thus, we still called this technique as SCRaMbLE.

3. Results: It could be a good idea to re-evaluate the labeling of the strains. More informative names (instead of long, unintelligible codes) would avoid confusion and make the manuscript more understandable. Perhaps most importantly, consider re-naming JDY525, as it is the strain that contains 83 Lox sites, and SparLox83 being the code name for the same strain that now also harbors the ReSCuES system. A more systematic, uniform naming would make it easier for the reader to follow. In addition, please indicate that JDY525 was constructed from strain BY4741 (this strain only pops up later in the text).

Response: Thanks for the reviewer’s great suggestion. JDY525 is the strain with 83 Sparsely distributed LoxPsym sites and SparLox83 is generated by integrating ReSCuES system into JDY525. To make the manuscript more understandable, we have modified JDY525 to SparLox83 and SparLox83 in the previous version to SparLox83R in the revised manuscript.

4. Results: Briefly explain the ReSCuES system.

Response: We appreciate the reviewer for the comments and we apologize for the confusing writing in the manuscript. We have explained the ReSCuES system in the revised manuscript as follows: “ReSCuES was a reporter to efficiently identify SCRaMbLed cells based on a loxP-mediated switch of “on” and “off” states of the two auxotrophic markers, *URA3* and *LEU2*.”

5. In the results sections, the authors mention statistical terms in the main text for observations for which no statistical tests were performed, for example: ‘no correlation’ (Extended Data Fig.5b), ‘significant’ (Extended Data Fig.3b, Extended Data Figs. 8b&c).

Response: Thanks for the reviewer to point this out. According to the reviewer’s suggestions, the correlation between the RW values and the distances of two loxPsym sites on different chromosomes were determined (Figure RL5a), as showed in revised Extended Data Fig. 7c.

As for Extended Data Fig.3b, the absence of data replication precluded the accurate application of a one-way ANOVA test. Thus, Pearson correlation coefficients (R package: psych, v2.3.9) were computed within the ARR of the four different lengths in haploid and diploid cells, yielding P values below 0.01 (Figure RL5b). Please see revised Extended Data Fig. 4b.

As for Extended Data Figs. 8b&c, we quantitatively assessed the growth of the strains before and after multiple rounds of directed evolution on YPD and YPD + HAc plates using CellProfiler v4.2.6. As the edges of the plates were relatively thick and had no neighboring colonies, providing a nutrient-rich environment and faster growth for the strains, we first removed the colonies located at the edges of the plates. For the remaining colonies, we utilized a customized yeast patch identification workflow (<https://cellprofiler.org/examples>) to quantify the mean intensity of their forced spots. Specifically, we first compressed the images into approximately 1200x800 png files. Then, we cropped them based on the positions of the colonies in the images. The "Block size" parameter in the CorrectIlluminationCalculate step was adjusted to 80, the "Typical diameter of objects" parameter in the IdentifyPrimaryObjects step was adjusted to 8-80, and the "Minimum FormFactor value" and "Minimum Area value" in the FilterObjects step were adjusted to 0.40 and 90, respectively. Finally, we performed an unpaired two-tailed t-test to assess the significance of the growth differences (Figure RL5c). Please see revised Extended Data Fig. 14b&c&d.

Figure RL5. Examples for statistical tests. **a.** Correlation between RW in haploid cells and 1/genomic distance between loxPsym sites on the different chromosome. **b.** ARR calculated using different flank

region lengths in haploid cells. Pearson correlation coefficients within the four lengths in haploid cells were analyzed. **c.** The growth of the strains on YPD of Round 3 were quantitatively evaluated using CellProfiler v4.2.6.

6. Summary: The claim that SparLox83xSynIII led to increased genomic rearrangements is not justified by the results; increased compared to which strain? SparLox83 and SynIII were not tested by themselves here.

Response: We appreciate the reviewer for the comments and we apologize for the confusing writing in the manuscript. To demonstrate that the loxPsym sites in SparLox83R could combine with synthetic chromosomes to generate genomic rearrangements, we detected the structural variations in nine SCRaMbLEd SparLox83RxSynIII colonies. Rearrangements were not only observed within SparLox83R-derived loxPsym sites and synIII-derived loxPsym sites, but also between them. Thus, we concluded that SparLox83RxSynIII led to increased genomic rearrangements compared with BY4741 × synIII, which was a strain with only synIII.

To avoid vagueness, we have revised the text as follows: “Moreover, SCRaMbLE of the hetero-diploid strains derived from crossing SparLox83R with strains possessing synthetic chromosome III (synIII) from the Sc2.0 project led to increased diversity of genomic rearrangements and relatively faster evolution of traits compared to a **hetero-diploid** strain with only synIII.”

7. Summary: “analysis of these evolved strains...”; needs rephrasing, the authors refer to the wrong experiment (SparLox83xSynIII instead of SparLox83).

Response: We appreciate the reviewer for the comments and sorry for the mistake. We have corrected the description as: “Analysis of **the SCRaMbLEd strain with increased tolerance to nocodazole** demonstrated that genomic rearrangements can perturb the transcriptome and 3D genome structure and can consequently impact phenotypes.”

8. Section 1: Construction of 83 genomic edits within one strain is very impressive and interesting. It would be very helpful to the field to gain insight in the number of unintended SNPs/InDels that were accumulated during strain construction.

Response: We appreciate the reviewer for the valuable suggestion, and we have indeed performed the SNPs/InDels analysis in SparLox83 using the short illumina reads derived from whole-genome shotgun sequencing. Totally, we identified 1197 SNPs/InDels in nuclear genome when we aligned the NGS reads to the yeast reference genome (GCF_000146045.2). These SNPs/InDels include 83 loxPsym insertion sites, a ReSCueS model, as well as the laboratory modifications when constructed the parental strain of SpaLox83. Furthermore, we detected 158 overlapping open reading frames (ORFs) associated with 305 SNPs/InDels. Notably, YAL063C (FLO9), YBR005W (RCR1), YBR148W (YSW1), and YEL042W (GDA1) exhibited significant frameshift mutations. The detailed information of SNPs/InDels is added in Table S3.

9. Section 1: It is confusing to me that strain JDY525 is used for these experiments, as the rest of the study is based on SparLox83. I suggest to add the results for this strain to Fig.1b.

Response: We appreciate the reviewer for the valuable suggestion. Initially, we employed a CRISPR-Cas9 based gene editing strategy to integrate loxPsym sites, resulting in the generation of JDY525 strain with 83 inserted sites. Given the addition of these 83 loxPsym sites, it was important

to evaluate the fitness and genome stability of the final strain. Subsequently, JDY525 was utilized to evaluate the phenotype on various types of media and perform PCRTAg analysis of 35 independent lineages after ~120 mitotic generations. After confirming that JDY525 remained unaffected in terms of genome stability and growth fitness, we integrated ReSCuES as a reporter into JDY525, generating SparLox83 strain to identify SCRaMbLEd cells. In the following SCRaMbLE experiments, we used SparLox83. We agreed with the reviewer that the phenotype of SparLox83 should also be contained in Fig. 1b. The serial dilution tests were performed as showed in the revised Fig. 1b.

Figure RL6. Revised Fig. 1b. SparLox83 and SparLox83R phenotyping on various media.

10. Section 1: Please add more information on the design of SparLox83: how were the insertion sites of LoxPsym sites chosen? Why were they not equally distributed amongst the chromosomes? Were they consistently inserted in 3'UTRs (as in the Sc 2.0 project) or also within ORFs/5'UTRs/...?

Response: We appreciate the reviewer's comments and we are sorry for the vague description. The principle of the loxPsym sites we chose is that these sites are located at the intergenic regions which contained a PAM site, NGG, and the insertions should not disrupt any possible functional elements. However, four off-target insertions were detected in the final strain, which were II-3, II-8, IV-8 and V-2. Three of the four sites (II-3, II-8, and V-2) were located at the ORFs of non-essential genes. As the strain did not show distinguishable growth defect, the three loxPsym sites were retained. The information on the design of SparLox83 was added in the revised text. Please see line 125-136 in page 7.

We have revised the text as follows: **“All the insertion sites were designed at the intergenic regions which contained a PAM site, NGG, without disrupting any possible functional elements.** Briefly, three loci were simultaneously targeted for integration of loxPsym sites in each cycle, eventually producing a strain with 83 sites (SparLox83) distributed across the 16 chromosomes, with at least two sites per chromosome (Fig. 1a). The presence of loxPsym at each locus was confirmed by whole genome sequencing (WGS; Table S1 and S2). **The final strain exhibited four off-target insertions, namely II-3, II-8, IV-8, and V-2. Among these sites, three (II-3, II-8, and V-2) were located within the ORFs of non-essential genes. Since the strain did not display any discernible growth defects (Fig. 1b), we opted to retain the three loxPsym sites. Meanwhile, multiple loxPsym sites were identified at the same loci through self-duplication of the plasmid backbone sequence (Fig. 1a and Extended Data Fig. 1b).”**

11. Fig.1a: Only 74 LoxPsym sites are depicted, please add all sites.

Response: We appreciate the reviewer's carefully reading and we are sorry for the vague description. According to the results of whole-genome sequencing, a sequence containing the sgRNA plasmid backbone sequence with two loxPsym sites was inserted into the targeted sites, including IV-12/IV-13, IX-2/IX-3, X-1/X-2, X-3/X-4 and XV-2/XV-3. Moreover, three duplications of the sgRNA

plasmid backbone sequence with four loxPsym sites were detected at XIII-1/XIII-2/XIII-3/XIII-4. Revisions have been made to Fig. 1a and specific descriptions have been incorporated into Extended Data Fig. 1b of the revised manuscript in order to provide a comprehensive presentation of these sites.

Figure RL7. Revised Fig. 1a and Extended Data Fig. 1b. Distribution of 83 loxPsym sites in the SparLox83 strain. **a.** A comprehensive overview of the 83 loxPsym sites across 16 chromosomes. **b.** The detailed information for the multiple sites at the same loci through self-duplication of the plasmid backbone sequence.

12. Fig. 1c: Could the authors provide the evidence for these claims as supplemental figure?

Response: According to the reviewer for the suggestion, we have supplemented the PCR verification results to Extended Data Fig. 3, which were amplified by using the genomic DNA of 35 independent colonies after ~120 mitotic generations as templates (Figure RL8).

Figure RL8. Revised Extended Data Fig. 3. PCR verification for 24 loxP sites in 35 independent colonies after ~120 mitotic generations. The absence of amplicons in the initial amplification was indicated by the red rectangles, while subsequent repetitions revealed the presence of correct amplicons, displayed at the bottom right corner.

13. Section 2: The authors stress that inter-chromosomal rearrangements pre-dominate over intra-chromosomal rearrangements. Although this is true for SparLox83, I wonder whether it is a fair general statement to make? In fact, intra-chromosomal rearrangements are more likely to occur (an observation which is also made in Fig. 2d and in section 6 (Table S6)), no?

Response: We appreciate the reviewer's question. Our sequencing analysis of the mixed culture indeed revealed a higher frequency of inter-chromosomal events compared to intra-chromosomal ones. However, we observed that there are more inter-chromosomal candidates than intra-

chromosomal candidates for each loxPsym site, potentially introducing bias. To accurately quantify the preference for rearrangements, it is necessary to normalize this bias. Thus, we introduced the Average Rearrangement Rate (ARR) to quantify the average rearrangement potential for each loxPsym site as showed in Fig. 2d. In the table S6, intra-chromosomal rearrangements were dominated because the events occurring between the loxPsym sites on synIII were included. The distances of synIII-derived sites were much closer than those of SparLox83R-derived sites, resulting a preference for interaction.

14. Section 3: It is unclear how ‘chromatin accessibility’ was defined/determined, this information should be added somewhere.

Response: Thanks for the reviewer’s suggestion. The definition of chromatin accessibility has been added to the revised manuscript as follows: “Chromatin accessibility refers to the degree physical accessibility and availability of DNA within the chromatin structure for interaction with various regulatory factors, such as transcription factors and other proteins. It describes the ability of these factors to bind specific genomic regions of the genome and regulate gene expression (*Nucleus.*, 2022, PMID: 36404679).” Please see line 248-253 in page 12.

15. Although I am convinced that the differences in recombination frequency that are observed between different LoxP sites are indeed mostly due to 1) their genomic distance and 2) the local chromatin structure, one cannot exclude other confounding factors, such as for example bias due to selection. Even in rich medium like YPD and optimal growth conditions, some rearrangements might cause a severe fitness defect or even cell death.... This could be pointed out more explicitly.

Response: Thank you for the reviewer’s constructive suggestion. We agree with the reviewer that it should be taken into consideration that the cells with different rearrangements varies in growth rate and bias may be introduced by phenotypic selection. However, it is difficult to distinguish whether the rearrangement preference is due to the fact that some pairs of loxPsym sites are prone to rearrange than other pairs or growth advantages brought about by rearrangements in some cells because both of them could result in increased cell amounts in the SCRaMbLEd pool for sequence. On the contrary, rearrangements which cause severe fitness defect may be covered in the cell pool as limited cells could spread in the population.

We have supplemented the discussion in the revision as: “Additionally, the distribution of cells with different rearrangements in the final population for sequence can be influenced by growth advantages or disadvantages resulting from structural variations. This factor also contributes to rearrangement preferences when utilizing the SCRaMbLEd pool for sequencing, although it is not easily distinguishable on its own.” Please see line 509-513 in page 22.

16. Fig. 2e: I suggest adding the information for intra-chromosomal ARR.

Response: Following the reviewer’s suggestion, the information for intra-chromosomal ARR has been added to Extended Data Fig. 6. As no significant correlation was observed for intra-chromosomal ARRs, this information was not provided in the previous manuscript.

Figure RL9. Revised Extended Data Fig. 6. Comparison of intra-chromosomal ARRs between loxPsym sites within (+) and outside (-) open chromatin regions in haploid cell.

17. Section 4: The data for growth on ethanol, H₂O₂ and NaCl are missing from Extended Data Fig. 6a, please provide this information.

Response: According to the reviewer's suggestion, the results have been added to Extended Data Fig. 8a of the revised manuscript.

Figure RL10. Revised Extended Data Fig. 8a. SCRaMbLED cells exhibited similar tolerance to the parent strain SparLox83R under stress conditions. Cells at log phase were 10-fold serially diluted onto YPD agar plates containing 10% ethanol, 3 mM H₂O₂, 1 M NaCl, or 3% glycerol and incubated for 2 days at 30°C.

18. Section 5: Could the authors analyze and describe the results in more detail? Information on how many recombination events were observed in total, how many per strain, how many inter-/intra-chromosomal and how many on synIII, how many of each type of recombination, is not mentioned, but valuable info. I believe some of this data is given in Table S6, but I would strongly encourage the authors to add an extra panel to Fig. 4 to discuss this information and adapt their interpretation of the results.

Response: According to the reviewer's suggestion, the status of each segment in the 9 strains were exhibited in the panel and the figure has been added to Extended Data Fig. 11b. In the figure,

recombination events were classified as deletions (gray), inversions (blue), duplications (yellow), duplicated inversion (dark blue), inverted duplication (light blue) and complex rearrangements (red) with respect to the parental synIII sequence according to previous report (*Genome Research*, 2016, PMID: 26566658). Based on this figure, we quantified and presented the counts of each type of recombination event on synIII in Table S5. Since it was challenging to determine whether adjacent segments underwent simultaneous rearrangement, we considered each segment as an individual unit for counting. Relevant information regarding the rearrangements of synIII has been included in the supplementary figures, as it was not deemed to be the focal point of this paper.

Figure RL11. Revised extended Data Fig. 11a and 11b. Rearrangements of synIII were observed in SCRaMbLE cells. **a.** Dot-plots illustrating synIII rearrangements in JDY553 and JDY596. Each dot represents a fragment between two loxPsym sites. Black dots represent fragments arranged in synIII sequence order and blue dots represent fragments inverted with respect to the synIII sequence. **b.** The fate of each segment in each strain was classified as deletion (gray), inversion (blue), duplication (yellow), duplicated inversion (dark green), inverted duplication (light green), multiple duplication (red) or identical to the parent strain (white).

19. Fig. 4a: Define y-axis, as the figure shows the structural rearrangements in synIII, rearrangements between synIII and chromosomes from SparLox83 should also be shown in this panel.

Response: We are sorry for the vague description. The figure 4a represented the segment order and orientation of rearranged synIII for the nine strains which were generated by SCRaMbLEgrams, a tool to depict the recombination events observed in SCRaMbLEd strains in Sc.2.0 project (*Genome Research*, 2016, PMID: 26566658). The y-axis indicates the name of a SCRaMbLEd strain and the x-axis was the arrangement of synIII which was visualized as a sequence of arrows. The arrows represent segments which are flanked by loxPsym sites on synIII. The color of each arrow indicates

the segment number in the parental chromosome, and the direction of the arrow represents the orientation.

There were two reasons that the rearrangement between synIII and chromosomes from SparLox83R was not shown in the figure. Firstly, the segment sizes flanked by two adjacent loxPsym sites of SparLox83R were significantly larger than those of synIII, resulting in distinct arrow representations for chrIII compared to synIII. The rearrangement between synIII and chrIII was an inter-chromosome event while the rearrangements displayed in Fig. 4a all belonged to intra-chromosomal events. Furthermore, a homologous recombination-mediated rearrangement occurred between synIII and chrIII, which was not mediated by loxPsym sites and could not be visualized as a sequence of arrows. Additionally, the comprehensive depiction of the rearrangements occurring between synIII and chrIII has been provided in Fig. 4c, encompassing all the detected rearrangements between synIII and chromosomes from SparLox83.

20. Fig. 5a: not very clear, please specify why some reads are grey/blue and add ChrA/B to all schematics.

Response: We apologize for the lack of clarity in our graphical representation. The color of ChrA has been modified from blue to gray, aligning it with the hue of the associated reads (1&2 in the Figure RL12). The four reads depicted in the figure correspond to the four types identified by sequencing: the reads 1 represents a sequence containing a loxPsym site from ChrA; the reads 2 represents a sequence without loxPsym sites from ChrA; the reads 3 represents a translocation mediated by a loxPsym site between ChrA and ChrB, as half of it could be mapped to ChrA and the remaining portion could be mapped to ChrB; the reads 4 represents a sequence without loxPsym sites from ChrB (the part within the dotted box). The copy number of each fragment of ChrA and ChrB can be determined by calculating based on the sequencing depth, which are represented as 4 reads in the figure. The modified figure has been updated in the revised manuscript.

Figure RL12. Revised Fig. 5a. The modified workflow of copy number variation analysis.

21. Fig. 5b: why is this data not shown for all chromosomes? Could the authors provide this information as supplemental material? And, add the data for ChrIII to the main figure?

Response: We apologize for the confusing writing. Indeed, the copy numbers of all loxPsym upstream and downstream regions have been assessed. A blue square represents a copy of the upstream or downstream region containing a loxPsym site, while a white square represents a region lacking such a site. Consequently, all regions of chromosomes could be represented by either a white square, or a blue square, or a combination. However, only a subset of regions on several chromosomes showed difference compared to the parent strain as presented in Fig. 5b. According to the reviewer's suggestion, the information of the rest chromosomes was provided as a supplemental figure, Extended Data Fig. 13.

As for chrIII, the presence of PCRTags in synIII enables the differentiation of reads from chrIII and synIII, thereby enabling to determinate their respective copy numbers. Additionally, the complex structural variations on synIII make it impossible to depict the upstream or downstream region of the loxPsym sites as presented in Fig. 5a. Moreover, the copy number variation of synIII has been illustrated in Fig. 4a and Extended Data Fig. 12. Therefore, only the data for ChrIII was provided in Extended Data Fig. 13.

Figure RL13. Revised Extended Data Fig. 13. Copy numbers of 2 kb upstream/downstream regions flanking the loxPsym sequence were determined for chromosomes without rearrangements in the nine diploid strains. The panels exhibited identical in all the nine strains but with varied synIII.

22. Section 7: not all populations show growth media with 0.4 % HAc (Extended Data Fig. 8b), please adapt the text.

Response: Thanks for pointing this out and we have modified the text as follow: “After the first round of SCRaMbLE (R1), **most of** the populations were able to grow on media with a maximum of 0.4% HAc (Extended Data Fig. 14b).”

23. Discussion: It is very interesting that more recombination events were observed in haploid strains compared to the diploid. However, the first explanation of the authors (that recombination in haploids might lead to lethality) actually suggests the opposite. The second explanation should be rephrased (it is not the same experiment) and could in principle be investigated (the data is available to look for LOH and aneuploidy in these strains). Could you hypothesize more on potential explanations and mention whether this has been observed previously?

Response: We are sorry for our pool writing for this section. In theory, more possibilities of recombination events could be detected in diploid cells because essential genes positioned between each pair of adjacent loxPsym sites would lead to the loss of viable cells and deletions are only occurred in diploid cells. Unexpectedly, we detected more rearrangement events detected in the haploid pool rather than in the diploid pool. Thus, we firstly speculated that the loss of loxPsym sites after SCRaMbLEing was the major reason. Therefore, we provided two possibilities to cause the loss of loxPsym sites. One was the deletion events which could only be occurred in diploid cells and the other was the LOH and aneuploidy events. According to the reviewer’s comment, we have rephrased the second explanation. Additionally, we provided another potential explanation and discussed the differences between our findings and those based on synthetic chromosomes.

The revised text was as follows: “Characterization of genetic rearrangements in SparLox83R and SparLox83R × BY4742 revealed that the proportion of rearrangement events was 10-fold higher in haploid than in diploid cells. There are several possible reasons behind this phenomenon. **First, the loss of loxPsym sites could only be occurred in diploid strains. In haploid strains, loxPsym-mediated deletion of fragments might lead to lethality because of essential gene loss^{47,48} while deletions can occur in the diploid without lethal effects because of the additional presence of the wild-type genome³⁴. Additionally, the possibility of the occurrence of LOH and aneuploidy**

events have been demonstrated by sequencing nine SCRaMbLEant colonies (Fig. 5). This might also be happened in the diploid strain of SparLox83R × BY4742 and further decrease the total number of loxPsym sites⁴⁹. Moreover, five rounds of induction-selection were performed before sequencing, and this may have further lowered the total number of loxPsym sites. Second, the additional copy of chromosome without loxPsym site may hinder the rearrangement capacity of the inter-chromosomal interaction. Although diploid colonies based on synthetic chromosomes of Sc2.0 exhibited a higher recombinational frequency compared to haploid colonies, no inter-chromosomal events were detected among the hundreds of sequenced diploid colonies³¹. We also detected a related lower inter-chromosomal interaction proportion in diploid strains (63.6%) than in haploid strains (84.8%) while the expected proportion is 7% for intra-chromosomal events (210/2775) and 93% for inter-chromosomal events (2565/2775) if there is no preference for intra- or inter-chromosomal recombination. Additionally, haploid strains may have high tolerance to gross inter-chromosomal events as its plasticity. However, in a diploid strain, recombination events between different chromosomes may give rise to two distinct genotypes of the homologous chromosome pairs, possibly causing genomic instability and severe growth defect⁵⁰. **More experimental validations are needed.**" Please see line 514-538 in page 22.

24. Discussion: I am confused about the mentioning of a haploid strain with an essential gene plasmid. I believe this was not mentioned in the result section, could you please clarify this?

Response: We are sorry for our careless that the literature was not cited in the manuscript. The strain harboring an essential gene plasmid was reported by Wang *et. al.* (PMID: 32268063). They put all essential genes of yeast chromosome III to a centromeric plasmid and transformed to the strain with synIII. Deletion frequencies of synIII chromosomal segments were revealed by investigation of SCRaMbLEd strains. Several segments with no deletion were detected in all strains, most of which were consistent with our results. Three segments were deleted in our strains which harboring a copy of wild type chrIII, but not in their strains.

To avoid misunderstanding, we have added the citation and modified the sentence as follows: "Segments 65, 66, and 67 were deleted in our strains but were not retained in the haploid strain **previous reported**³⁰, which was possessed an essential gene plasmid, indicating that synthetic lethal interactions can be bypassed when a copy of wild type chrIII is present."

25. Discussion: At the end of the section, the authors mention that an increased frequency of genomic rearrangements between SparLox83-derived chromosomes and synthetic chromosomes can be expected. I wonder why this is claimed, as the data in the manuscript suggests otherwise (e.g. the general decrease in genomic rearrangements for diploid strains (fig.1) and the low frequency of SparLox83 rearrangements in the presence of other synthetic chromosomes (Table S6))?

Response: We apologize for the confusion and thank you for the comment. The SCRaMbLE system is firstly applied for synthetic chromosomes. Several previous studies have revealed that there is a preference for intra-chromosome recombination than the inter-chromosome recombination and the potential of SCRaMbLE to study genomic rearrangements is currently hindered. Here, a strain with 83 loxPsym sites distributed across the 16 chromosomes was constructed. A variety of large-scale genomic rearrangements, especially inter-chromosomal events, were detected in the resultant SCRaMbLEd yeast population. In addition, we demonstrated that the large-scale genomic

rearrangements could generate advantageous phenotypes and the loxPsym sites can interact with those from synthetic chromosome. Although decreased genomic rearrangements were identified in diploid strains (Fig. 2), more rapid environmental adaptation was observed in diploid strains than in haploid strains (Figure RL3) and LOH and aneuploidy were detected in single-colony sequencing which were missed in population sequencing. Similarly, strains harboring SparLox83-derived chromosomes and synthetic chromosomes showed an accelerated gain of HAc tolerance (Fig. 6). Our results indicated that diploid strains with SparLox83-derived loxPsym sites and synthetic chromosomes can generate more structural variations than haploid strains and diploid strains with only synthetic chromosomes when exposure to the stress because genotypic diversity is an important driver of adaptation to new environments. In addition, a translocation was observed between chromosome III and synIII (Fig. 4c). If the strain with 16 synthetic chromosomes is available, the possibility that interactions between SparLox83-derived chromosomes and synthetic chromosomes would be increased as two copies of any chromosomes harbor loxPsym sites. Based on these, we speculate that the frequency of translocation events using the final Sc2.0 yeast strains harboring all synthetic chromosomes would be increased.

26. Discussion: you mention that chromosomal rearrangements (or more commonly: structural variation) is a known, important driver of adaptive evolution. This is especially the case in yeast, where genomic rearrangements are often hypothesized to offer a “quick fix” to a sudden, harsh challenge (as they are often used in ALE). It would perhaps be appropriate to cite some of the pioneering work here, and discuss in more depth how SCRaMbLE could be used to study this process in more detail.

Response: Thank you for the suggestion. According to the reviewer’s advice, we have further discussed this in the discussion section of the revised version as follows:

“Previous studies have demonstrated the potential of SCRaMbLE for directed evolution^{24,26,27}. The SCRaMbLE experiment typically involves inducing Cre recombinase in yeast cells containing one or more synthetic chromosomes to generate a population with genomic heterogeneity. This population is then subjected to selective growth conditions for the screening of viable cells⁵². Compared with classical adaptive laboratory evolution via causal mutations, SCRaMbLE facilitates rapid and extensive genetic rearrangements, including deletion, inversion and duplication. Additionally, aneuploidy is also considered as a quick response to stress⁵³. Our findings suggested that SCRaMbLE based on the loxPsym sites across the 16 chromosomes in SparLox83R led to the potential occurrence of aneuploidy, such as whole chromosome deletion and duplication (Fig. 5c), which were rarely observed in cells with synthetic chromosomes. Thus, these two types of SCRaMbLE could complement each other and be orthogonal to other directed evolution methods as they provide different types of genetic changes⁵⁴.” Please see line 580-593 in page 24.

27. Discussion: Please discuss if and how you could overcome the huge preference for recombination between proximal Lox sites in the complete Sc2.0? If this is not resolved, the repertoire of rearrangements that will result from recombining the SCRaMbLE system in Sc2.0 will be quite limited?

Response: Thank you for the suggestion. The SCRaMbLE system in Sc2.0 is designed to generate population-level diversity in yeast strains through the utilization of the Cre/loxPsym system. The synthetic chromosomes facilitate large-scale genomic rearrangements between any two loxPsym

sites within or across them. Recent studies investigating SCRaMbLE without selection pressure have highlighted the significant influence of Cre enzyme abundance, genome ploidy, and chromatin structures on recombination frequencies and resulting outcomes. These findings suggested there is a preference for interaction with specific loxPsym sites. However, when employing SCRaMbLE as part of a directed evolution strategy, it typically requires exposing yeast cells to challenging environments such as selective pressures or phenotypic screening to identify cells with beneficial genomic rearrangements that enhance adaptation to environmental stress over multiple generations. Furthermore, while rearrangement events are more likely to occur at short distances, long-range recombination events have also been observed. This can be attributed partially to multiple recombination events undergone by SCRaMbLEd genomes (*bioRxiv*, ID: 507906). Additionally, SCRaMbLE has the capability to reshape both content and order of chromosomes. Thus, each round of SCRaMbLE introduces different proximal loxPsym site configurations which may help to overcome inherent recombination preferences.

According to the reviewer's suggestion, the related discussion have been added: "Although limited interactions were observed between SparLox83-derived chromosomes and synthetic chromosomes, strains harboring two genotypes of chromosomes exhibited an accelerated acquisition of HAc tolerance (Fig. 6). These findings suggested that exposure to challenging environments was crucial for identifying cells with genomic rearrangements when utilizing the SCRaMbLE system in directed evolution experiments. The composition and arrangement of chromosomes can be reshaped, resulting in different interaction preferences within loxPsym sites during each round of SCRaMbLE. By increasing the number of SCRaMbLE rounds, variants with enhanced fitness can be enriched." Please see line 570-578 in page 24.

28. Methods, yeast colony PCR: indicate polymerase

Response: Thanks for pointing this out and the polymerase used for yeast colony PCR have been indicated.

29. Methods, SCRaMbLE: provide information on the Cre-plasmid (Addgene ID/reference), specify if cells were kept in liquid cultures for consecutive rounds of SCRaMbLE or if they were plated in between.

Response: The plasmid was constructed by integrating the NatMX cassette into the LYS2 coding sequence based on the plasmid reported by Luo et. al. (*Genome Biology*, 2021, PMID: 33397424). For the consecutive rounds, cells were kept in liquid cultures during SCRaMbLE. We have modified the related information in the revised manuscript. Please see line 893-895 in page 45.

"SparLox83R or the heterozygous diploids with ReSCuES sequence integrated were transformed with pGAL-Cre plasmid, which was constructed by integrating the NatMX cassette into the LYS2 coding sequence based on a previously described plasmid²¹."

30. Data availability statement is missing and access of the WGS data is not provided.

Response: Thanks for pointing this out and we have supplemented data availability statement in the revised manuscript. Please see line 1041-1048 in page 50.

31. Table S2: specify the locus of ReSCuES integration in SparLox83.

Response: We appreciate the reviewer for pointing this out and we have added the information to table S2.

Typos:

1. Summary: heterodiploid strains < strain; strains possessing synIII < strain

Response: This has been corrected.

2. Results, section 2: $r=0.9828$ >< fig.2c mentions R2

Response: This has been corrected.

3. Results, section 4: reference Luo et al., 2018 < reference 26

Response: This has been corrected.

4. Results, section 5: ‘often’ deleted: often should be removed if all 9 colonies were obtained from a single induced population. It should be mentioned clearly in the text how many biological repeats were used. If this was one biological repeat, also the discussion ‘similar biases’ should be adapted.

Response: According to the reviewer’s suggestion, ‘often deleted’ has been revised to “prone to delete”. “Similar biases” has been revised to “high similarity”.

5. Results, Fig.5: bule < blue

Response: This has been corrected.

6. Results, section 5: mention LOH=loss of heterozygosity

Response: This has been corrected.

7. Discussion, were existed < existed

Response: This has been corrected.

8. Methods, SCRaMbLE: SparL83<SparLox83

Response: This has been corrected.

9. Methods, Split-URA3: palte<plate, inducted<induced, other typo’s

Response: This has been corrected.

10. Methods, WGS: according<according

Response: This has been corrected.

11. Methods, junction detection: number 2 seems to be a mistake in formula $N_{rei}/2N_{ormi}+N_{rei}$

Response: We thank the reviewer for the careful reading. We have confirmed that the “number 2” was right.

12. Methods, Hi-C library: Spar83L<SparLox83

Response: This has been corrected.

Reviewers' Comments:

Reviewer #2:

Remarks to the Author:

Dear authors,

In general, I am very happy to see how you addressed the comments of myself as well as the other reviewer. I believe that these adaptations have made an already impressive manuscript even stronger and, above all, more accurate. As such, I support publication.

That said, I am less satisfied by the response to my very first comment, namely that no "no-recombination / no-SCRAMBLE" controls have been used in this manuscript. I understand that a previous SCRAMBLE paper has reported these controls, but this is not a good reason to not include the proper controls in this new study, which uses a different SCRAMBLE setup. I would ask the authors to at least mention explicitly that their data do not prove that the SCRAMBLEing leads to quicker adaptation. It is of course OK to also mention that the results reported in PMID:29789540 do suggest that SCRAMBLE is indeed essential.

kind regards,
Kevin Verstrepen

REVIEWERS' COMMENTS

Reviewer #2 (Remarks to the Author):

Dear authors,

In general, I am very happy to see how you addressed the comments of myself as well as the other reviewer. I believe that these adaptations have made an already impressive manuscript even stronger and, above all, more accurate. As such, I support publication.

That said, I am less satisfied by the response to my very first comment, namely that no "no-recombination / no-SCRAMBLE" controls have been used in this manuscript. I understand that a previous SCRAMBLE paper has reported these controls, but this is not a good reason to not include the proper controls in this new study, which uses a different SCRAMBLE setup. I would ask the authors to at least mention explicitly that their data do not prove that the SCRAMBLEing leads to quicker adaptation. It is of course OK to also mention that the results reported in PMID:29789540 do suggest that SCRAMBLE is indeed essential.

kind regards,

Kevin Verstrepen

Response: Thank the reviewer for their recognition of our work. According to the reviewer's suggestion, we have supplied the statement that our data did not prove quicker adaptation by SCRaMbLE in the revised discussion section.

The revised text is as follows: "However, further investigation is necessary to determine whether SCRaMbLE could provide a more rapid response to stress than classical adaptive laboratory evolution." Please see page 25, line 591-593.